# DHP: DISCRETE HIERARCHICAL PLANNING FOR HRL AGENTS

## ABSTRACT

Hierarchical Reinforcement Learning (HRL) agents often struggle with long-horizon visual planning due to their reliance on error-prone distance metrics. We propose Discrete Hierarchical Planning (DHP), a method that replaces continuous distance estimates with discrete reachability checks to evaluate subgoal feasibility. DHP recursively constructs tree-structured plans by decomposing long-term goals into sequences of simpler subtasks, using a novel advantage estimation strategy that inherently rewards shorter plans and generalizes beyond training depths. In addition, to address the data efficiency challenge, we introduce an exploration strategy that generates targeted training examples for the planning modules without needing expert data. Experiments in 25-room navigation environments demonstrate a $100\%$ success rate (vs. $90\%$ baseline). The method also generalizes to momentum-based control tasks and requires only $\log N$ steps for replanning. Theoretical analysis and ablations validate our design choices.

## 1 INTRODUCTION

Hierarchical planning enables agents to solve complex tasks through recursive decomposition Dietterich (2000), but existing approaches face fundamental limitations in long-horizon visual domains. While methods using temporal distance metrics Pertsch et al. (2020); Ao et al. (2021) or graph search Eysenbach et al. (2019) have shown promise, their reliance on precise continuous distance estimation creates two key challenges:

- **Coupled Learning Dynamics**: Distance estimates depend on the current policy's quality, where suboptimal policies produce misleading distances and practical implementations may require arbitrary distance cutoffs Ao et al. (2021); Eysenbach et al. (2019).

- **Exploratory Objective**: The intrinsic rewards used for training explorers may not align with the planning objective leading to inaccurate distance measures by design (Fig. 9).

We address these issues by reformulating hierarchical planning through discrete reachability – a paradigm shift from "How far?" to "Can I get there?". Our method (DHP) evaluates plan feasibility through binary reachability checks rather than continuous distance minimization. This approach builds on two insights: local state transitions are easier to model than global distance metrics, and reachability naturally handles disconnected states through $0/1$ signaling.

Our key contributions:

- A reachability-based discrete reward scheme for planning. (Sec. 2.3.2, Fig. 7d).

- An advantage estimation algorithm for tree-structured plans (Sec. 2.3.3, Fig. 7b).

- A memory conditioned exploration strategy outperforming expert data (Sec. 2.4, Figs. 7c,10).

- An extensive ablation study that measures the contribution of each module (Sec. 3.2).

Empirical results confirm these design choices. On the 25-room navigation benchmark, DHP achieves perfect success rates ($100\%$ vs $90\%$) with shorter path lengths (Sec. 3.1). Our ablation studies demonstrate that both the reachability paradigm and the novel advantage estimation contribute significantly to these gains (Section 3.2). We also test our method in a momentum-based RoboYoga environment for generalizability. Video: `https://sites.google.com/view/dhp-video/home`.

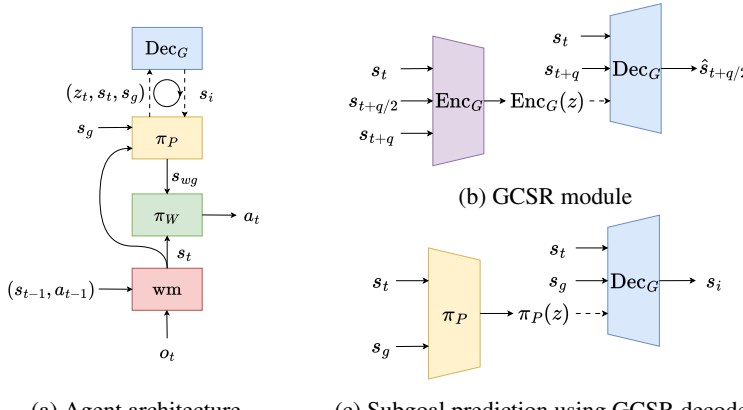

(a) Agent architecture    (c) Subgoal prediction using GCSR decoder

Figure 1: Illustrations for different module architectures. (a) Overall planning agent architecture. The world model predicts the state $s_t$, the planner takes the current and goal states $(s_t, s_g)$ to output a latent variable $z$, the GCSR Decoder is then used to predict a subgoal $s_i$. Then the subgoal is used as a goal to predict another subgoal. This continues recursively till a reachable subgoal $s_{wg}$ is found, which is then passed to the worker. (b) The GCSR module is a conditional VAE that consists of an encoder and a decoder optimized to predict midway states, given the initial and final states. (c) The planning policy uses the GCSR decoder to predict subgoals.

## 2 DISCRETE HIERARCHICAL PLANNING (DHP)

In the context of Markovian Decision Processes (MDP), a Reinforcement Learning task can be imagined as an agent transitioning through states $s_t$ using actions $a_t$. Our task is to find the shortest path between any two given states $(s_t, s_g)$. To do this, we first use an explorer to collect a dataset of useful trajectories possible within the environment. Then we learn a planning policy that learns to predict subgoals, decomposing the initial task into two simpler subtasks. The recursive application of the policy breaks the subtasks further till subtasks directly manageable by the worker are found.

### 2.1 AGENT ARCHITECTURE

We use the base architecture from the Director Hafner et al. (2022) as it has been observed to provide a practical method for learning hierarchical behaviors directly from pixels. It consists of three modules: world-model, worker, and manager. The world model is implemented using the Recurrent State Space Module (RSSM) Hafner et al. (2019), which learns state representations $s_t$ using a sequence of image observations $o_t$. The worker policy $\pi_W$ learns to output environmental actions $a_t$ to reach nearby states $s_{wg}$. The manager $\pi_M$ is a higher-level policy that learns to output desirable sub-goal states $s_{wg}$ for the worker (updated every $K$ steps) in the context of an external task and the exploratory objective jointly. We replace the manager with our explorer policy $\pi_E$ and the planning policy $\pi_P$ as required. Figure 1a shows the agent architecture during inference using $\pi_P$. The worker and the world-model are trained using the default objectives from the Director (for more details see Appendix B). First, we describe our planning policy and then derive an exploratory strategy fit for it.

### 2.2 PLANNING POLICY

The Planning policy $\pi_P$ is a goal-conditioned policy that takes the current and goal state as inputs to yield a subgoal. However, predicting directly in the continuous state space leads to the problem of high-dimensional continuous control Hafner et al. (2022); Pertsch et al. (2020). To reduce the search space for the planning policy, we train a discrete Conditional Variational AutoEncoder (CVAE) that learns to predict midway states $s_{t+q/2}$ given an initial and a final state $(s_t, s_{t+q})$. We refer to this module as Goal-Conditioned State Recall (GCSR). It consists of an Encoder and a Decoder (Fig. 1b). The Encoder takes the initial, midway, and final states $(s_t, s_{t+q/2}, s_{t+q})$ as input to output a distribution over a latent variable $\text{Enc}_G(z|s_t, s_{t+q/2}, s_{t+q})$. The Decoder uses the states $(s_t, s_{t+q})$ and a sample from the latent distribution $z \sim \text{Enc}_G(z)$ to predict the midway state $\text{Dec}_G(s_t, s_{t+q}, z) \rightarrow \hat{s}_{t+q/2}$. The module is optimized using the data collected by the explorer to minimize the ELBO objective (Eq. 1). A mixture of categoricals $(4 \times 4)$ is used as the prior latent distribution $p_G(z)$ for all our experiments. Triplets at multiple temporal resolutions $q \in Q$ are

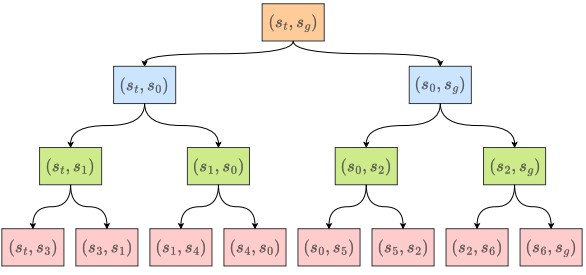
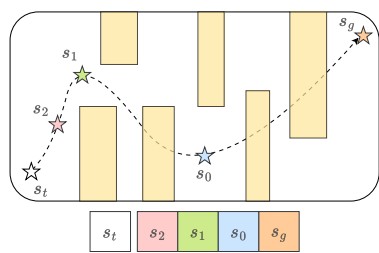

(a) Unrolled subtask tree during Training

(b) Planning during Inference

Figure 2: The figure illustrates the plan unrolling process during the training and inference phases. (a) During **Training**, the initial task $(s_t, s_g)$ is recursively decomposed into smaller tasks by midway subgoal prediction to generate a subtask tree. The lowest level nodes represent the simplest decomposition of the initial task as: $(s_t, s_g) \rightarrow (s_t, s_3, s_1, s_4, s_0, s_5, s_2, s_6, s_g)$. (b) During **Inference**, only the first branch of the tree is unrolled. Here, the agent is tasked with reaching $s_g$. So it first divides the task $(s_t, s_g)$ into two chunks by inserting $s_0$. Then it proceeds to divide the subtask $(s_t, s_0)$ by inserting $s_1$ and ignoring the second part $(s_0, s_g)$. The recursive division continues till the first subgoal reachable in $K$ steps is found. This results in a stack of subgoals shown at the bottom.

extracted, allowing the agent to function at all temporal resolutions. For minimizing overlap between resolutions, we use exponentially increasing temporal resolutions, $Q = \{2K, 4K, 8K, ...\}$.

$$\mathcal{L}(\text{Enc}_G, \text{Dec}_G) = \sum_{q \in Q} \left\| s_{t+q/2} - \text{Dec}_G(s_t, s_{t+q}, z) \right\|^2 + \beta \text{KL}[\text{Enc}_G(z|s_t, s_{t+q/2}, s_{t+q}) \parallel p_G(z)]$$

$$\text{where} \quad z \sim \text{Enc}_G(z|s_t, s_{t+q/2}, s_{t+q}) \tag{1}$$

The planning policy predicts in the learned latent space, which are expanded into sub-goals using the GCSR decoder as: $\text{Dec}_G(s_t, s_g, z)$ where $z \sim \pi_P(z|s_t, s_g)$ (Fig. 1c). Note that the word *discrete* in our title refers to our plan evaluation method (Sec. 2.3.2), not the action space of the planning policy.

## 2.3 PLANNING POLICY OPTIMIZATION

The planning policy is optimized as a Soft-Actor-Critic (SAC) Haarnoja et al. (2018) in three steps: construct plans between random state pairs (Sec. 2.3.1), plan evaluation using discrete rewards and our novel advantage estimation method (Sec. 2.3.2, 2.3.3), and policy updates using policy gradients (Sec. 2.3.4).

### 2.3.1 PLAN UNROLLING

Given the initial and final states $(s_t, s_g)$, subgoal generation methods predict an intermediate subgoal $s_0$ that breaks the task into two simpler subtasks $(s_t, s_0)$ and $(s_0, s_g)$. The recursive application of the subgoal operator further breaks the task, leading to a tree of subtasks $\tau$. The root node $n_0$ represents the original task $(s_t, s_g)$, and each remaining node $n_i$ in the tree $\tau$ represents a sub-task. At each node $n_i$, the policy predicts a subgoal as: $s_i = \text{Dec}_G(n_{i,0}, n_{i,1}, z)$ where $z \sim \pi_P(z_i|n_i)$. The preorder traversal of the subtask tree of depth $D$ can be written as $n_0, n_1, n_2, ..., n_{2^{D+1}-2}$. Figure 2a shows an example unrolled tree. The lowest-level nodes show the smallest decompositions of the task under the current planning policy $\pi_P$.

**Inference:** Unlike traditional methods for hierarchical planning (eg, cross-entropic methods (CEM) Rubinstein and Kroese (2004); Nagabandi et al. (2018)), which require unrolling multiple full trees followed by evaluation at runtime, our method does not require full tree expansion. A learned policy always predicts the best estimated subgoal by default. Thus, we can unroll only the leftmost branch, as only the first reachable subgoal is required (Fig. 2b). Efficient planning allows the agent to re-plan at every goal refresh step $(K)$, thereby tackling dynamic and stochastic environments.

### 2.3.2 DISCRETE REWARDS SCHEME

We want to encourage trees that end in subtasks manageable by the worker. A subtask is worker-manageable if the node goal $n_{i,1}$ is reachable by the worker from the node initial state $n_{i,0}$. Since learning modules inside other learned modules can compound errors, we use a more straightforward method to check reachability. We simulate the worker for $K$ steps using RSSM imagination, initialized at $n_{i,0}$ with goal $n_{i,1}$. If the `cosine_max` similarity measure between the worker's final state $s_{t,i}$ and the assigned goal state is above a threshold $\Delta_R$, the node is marked as *terminal* (Eq. 2). The *terminal* nodes do not need further expansion, which is different from the word's usual meaning in the context of trees. The lowest layer non-*terminal* nodes in a finite-depth unrolled tree are called *truncated* nodes. Computing the *terminal* array $T_i$ allows supporting imperfect trees with branches terminating at different depths. A plan is considered *valid* if all its branches end in *terminal* nodes. The policy is rewarded 1 at *terminal* nodes and 0 otherwise (Eq. 3). A discrete reward scheme enables optimization that increases the likelihood of *valid* plans compared to distance-based approaches, which gradually optimize the policy to reduce path length.

$$T_i = T_{(i-1)/2} \vee \text{cosine\_max}(s_{t,i}, n_{i,1}) > \Delta_R \tag{2}$$

$$R_i = \begin{cases} 1, & \text{if } T_i == True \\ 0, & \text{otherwise} \end{cases} \tag{3}$$

### 2.3.3 RETURN ESTIMATION FOR TREES

Taking inspiration from the standard discounted return estimation for *linear* trajectories (sequence of states) Sutton and Barto (2018), we propose an approach for tree trajectories. Returns for a *linear* trajectory are computed as the reward received at the next step and discounted rewards thereafter, $G_t = R_{t+1} + \gamma G_{t+1}$. Similarly, the return estimate for trees is the minimum discounted return from the child nodes. Given a tree trajectory $\tau$, the Monte-Carlo return (Eq. 4), the 1-step return (Eq. 5), and the lambda return (Eq. 6) for each non-*terminal* node as can be written as:

$$G_i = (1 - T_i) \cdot \min(R_{2i+1} + \gamma G_{2i+1}, R_{2i+2} + \gamma G_{2i+2}) \tag{4}$$

$$G_i^0 = (1 - T_i) \cdot \min(R_{2i+1} + \gamma v_P(n_{2i+1}), R_{2i+2} + \gamma v_P(n_{2i+2})) \tag{5}$$

$$G_i^\lambda = (1 - T_i) \cdot \min(R_{2i+1} + \gamma((1 - \lambda)v_P(n_{2i+1}) + \lambda G_{2i+1}^\lambda), R_{2i+2} + \gamma((1 - \lambda)v_P(n_{2i+2}) + \lambda G_{2i+2}^\lambda)) \tag{6}$$

All branches should end in *terminal* nodes to score a high return with the above formulation. Additionally, since the discount factor diminishes the return with each additional depth, the agent can score higher when the constructed tree is shallow (less maximum depth). This characteristic is similar to *linear* trajectories, where the return is higher for shorter paths to the goal Tamar et al. (2016).

Fig. 3 illustrates example return evaluations for Monte-Carlo returns and $n$-step truncated returns that use value estimates to replace rewards at the non-*terminal truncated* nodes. The $n$-step returns allow for generalization beyond the maximum unrolled depth $D$ (Sec. A.2.2). Thus, the tree can be unrolled for higher depths $D_I$ during inference. We show that the Bellman operators for the above returns are contractions and repeated applications cause the value function $v_P^\pi$ to approach a stationary $v^*$ (Sec. A.4). We explore how the return penalizes maximum tree depth and encourages balanced trees (Sec. A.2.3), implying that the optimal policy inherently breaks tasks roughly halfway.

### 2.3.4 POLICY GRADIENTS

Using tree return estimates (we use $n$-step lambda returns in all cases), we derive the policy gradients for the planning policy as (Eq. 7, proof in Sec. A.1). We show that if the function $G_i^\lambda$ is replaced by a function independent of the policy actions, the expectation reduces to 0, implying that we can use the value function as a baseline for variance reduction (Th. A.2). With policy gradients and an entropy term, to encourage random exploration before convergence, we construct the loss function for the actor $\pi_P$ and critic $v_P$ as (sum over all nodes except the *terminal* and *truncated* nodes):

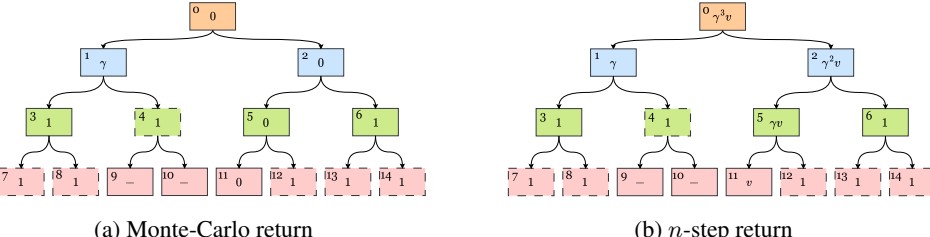

(a) Monte-Carlo return                                    (b) $n$-step return

Figure 3: Example return estimations for an imperfect tree (node indices at top-left). The dash-bordered cells indicate *terminal* nodes where the policy receives a 1 reward and the branch terminates. While one of the branches terminates early ($i = 4$), one does not for the unrolled depth ($i = 11$). (a) Since we compute the return as the $\min$ of the child nodes, the Monte-Carlo return at the root node is 0 in this case. However, a positive learning signal is still induced at nodes ($i = 1, 3, 6$). (b) Using the critic as a baseline for computing $n$-step returns. The $n$-step returns allow bootstrapping by substituting the reward with value estimates $v$ at the *truncated* node ($i = 11$). This induces a learning signal at the root node even if the plan is incomplete for the unrolled depth.

$$\nabla_{\pi_P} J(\pi_P) = \mathbb{E}_\tau \sum_{i=0}^{2^D - 2} G_i^\lambda(\tau) \nabla_{\pi_P} \log \pi_P(z_i | n_i) \tag{7}$$

$$\mathcal{L}(\pi_P) = -\mathbb{E}_{\tau \sim \pi_P} \sum_{i=0}^{2^D - 2} (1 - T_i) \cdot (G_i^\lambda - v_P(n_i)) \log \pi_P(z_i | n_i) + \eta \mathbb{H}[\pi_P(z_i | n_i)] \tag{8}$$

$$\mathcal{L}(v_P) = \mathbb{E}_{\tau \sim \pi_P} \sum_{i=0}^{2^D - 2} (1 - T_i) \cdot (v_P(n_i) - G_i^\lambda)^2 \tag{9}$$

## 2.4 EXPLORER

During exploration, an exploration policy $\pi_E$ is used as a manager to drive the worker behavior. For the same problem of continuous control, the Explorer predicts goals in a discrete latent space learned using a VAE. As the predicted states do not need to be conditioned on other states like the planner, the Explorer VAE learns state representations unconditionally (similar to Director). The Explorer VAE consists of an encoder and a decoder ($\text{Enc}_U, \text{Dec}_U$). The encoder predicts latent distributions using state representations: $\text{Enc}_U(z | s_t)$, and the decoder tries to reconstruct the states using the samples from the latent distribution: $\text{Dec}_U(z)$ (Fig. 4a). As the prediction space is unconstrained, we use a larger latent size ($8 \times 8$ mixture of categoricals). The VAE is optimized using the ELBO loss (Eq. 25). The Explorer is implemented as an SAC (Fig. 4b) and optimizes an intrinsic exploratory reward.

### 2.4.1 EXPLORATORY REWARD

Since the planning policy $\pi_p$ uses the GCSR decoder for subgoal prediction, it is limited to the subgoals learned by the GCSR module. The GCSR training data must contain all possible state triplets in the environment for the planner to function well. To achieve this, we formulate our exploratory objective to encourage the Explorer to enact trajectories that are not well-modeled by the GCSR module. If Explorer traverses a trajectory that contains a state triplet $(s_t, s_{t+q/2}, s_{t+q})$, the modeling error is measured as the mean squared error between sub-goal $s_{t+q/2}$ and its prediction using the GCSR decoder. We compute the exploratory rewards for state $s_t$ based on the previous states as:

$$R_t^E = \sum_{q \in Q} \left\| s_{t-q/2} - \text{Dec}_G(s_{t-q}, s_t, z) \right\|^2 \quad \text{where} \quad z \sim \text{Enc}_G(s_{t-q}, s_{t-q/2}, s_t) \tag{10}$$

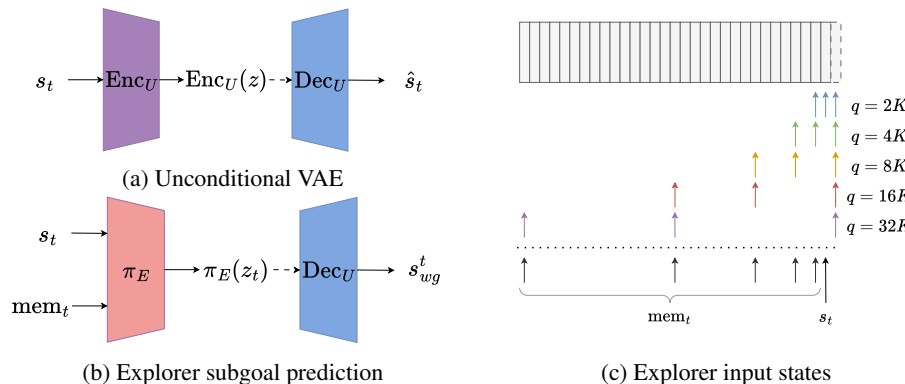

(a) Unconditional VAE

(b) Explorer subgoal prediction

(c) Explorer input states

Figure 4: (a) Unconditional VAE that learns to predict states not conditioned on other states. (b) The explorer uses the memory and the VAE decoder to predict subgoals for the worker. (c) Illustration showing the states required as inputs for the Explorer for an example trajectory (top) being played out by an agent. It is a coarse trajectory that shows every $K$-th frame. The agent is at state $s_t$ and will receive rewards when it moves into the placeholder future state $s_{t+1}$ (dashed border). The rewards at $s_{t+1}$ will be computed using the GCSR for different temporal resolutions $q \in Q$ indicated on the right. The colored arrows indicate the state triplet required to compute the exploratory reward at $s_{t+1}$. Combining these state dependencies and removing redundancies yields the input requirements indicated below the dashed line. The inputs consist of the current state and the memory.

### 2.4.2 MEMORY AUGMENTED EXPLORER

Since the exploratory rewards for the current step depend on the past states, the Explorer needs to know them to guide the agent accurately along rewarding trajectories. Fig. 4c shows the state dependencies for the exploratory rewards at different temporal resolutions $q$. Each temporal resolution $q$ requires $(s_{t-q}, s_{t-q/2})$ for computing reward. For our case, the past states required are: $[(s_{t-K}, s_{t-2K}), (s_{t-2K}, s_{t-4K}), ...]$ for each $q \in \{2K, 4K, ...\}$ respectively. Removing the duplicates reduces the set of states required to: $\{s_t, s_{t-K}, s_{t-2K}, ...\}$.

To address this, we provide a memory of the past states as an additional input to the exploratory manager SAC $(\pi_E(s_t, \text{mem}_t), v_E(s_t, \text{mem}_t))$ (Fig. 4b). We implement this by maintaining a memory buffer that remembers every $K$-th visited state. Then, we extract the required states as memory for the Explorer. For a trajectory rollout of length $T$, the required size of the memory buffer is $L_{\text{mem}} = T/K$, and the size of the memory input is $\log_2 L_{\text{mem}}$.

**Practical Consideration:** It can be practically infeasible to maintain a large memory buffer. However, our memory formulation is highly flexible and allows us to ignore exploratory rewards that require states far in the past. For all our experiments, we use $T = 64$ length rollouts with $L_{\text{mem}} = 8$ and memory input size: 3. The trajectory length also limits the temporal resolutions $q$ for which the exploratory rewards can be computed. While not entirely optimal, this is sufficient for a significant performance improvement (Fig. 7c).

### 2.4.3 POLICY OPTIMIZATION

The Explorer is optimized as an SAC using the policy gradients from the REINFORCE objective. RSSM is used to imagine trajectories $\kappa$, followed by extracting every $K$-th frame of the rollout as abstract trajectories. For clarity, let the time steps of the abstract trajectory be indexed by $k$, where $k = t/K$. Then, the lambda returns $G_k^\lambda$ are computed for the abstract trajectories using a discount factor $\gamma_L$ (Eq. 11). Finally, the lambda returns are used to formulate the explorer actor and critic loss (Eq. 12,13).

(a)                                                                                          (b)

Figure 6: Samples plan by our agent during training and inference. (a) Samples of full plans as sequences of subgoals from the start states (left) to the goal states (right). The subgoals are extracted from the goal states of all *terminal* nodes. The blanks occur when nodes terminate before maximum depth. (b) Sample subgoals generated during inference. The first and last images indicate the initial and goal states. Other images represent subgoals that break the path from the initial to the subgoal image on its right [Same order as 2b]. Blanks are when a reachable subgoal is found before the max depth.

$$G_k^\lambda = R_{k+1}^E + \gamma_L((1-\lambda)v_E(s_{k+1}) + \lambda G_{k+1}^\lambda) \tag{11}$$

$$\mathcal{L}(\pi_E) = -\mathbb{E}_{\kappa \sim \pi_E} \sum_{k=0}^{T/K-1} (G_k^\lambda - v_E(s_k, \mathrm{mem}_k)) \log \pi_E(z_k|s_k, \mathrm{mem}_k) + \eta\mathbb{H}[\pi_E(z_k|s_k, \mathrm{mem}_k)] \tag{12}$$

$$\mathcal{L}(v_E) = \mathbb{E}_{\kappa \sim \pi_E} \sum_{k=0}^{T/K-1} (v_E(s_k, \mathrm{mem}_k) - G_k^\lambda)^2 \tag{13}$$

## 3 EVALUATION & RESULTS

**Task Details**

We extensively test our agent in the 25-room environment, a 2D maze task where the agent must navigate through connected rooms to reach the target position. Benchmarks from previous methods show average episode duration $> 150$ steps, indicating a long-horizon task Pertsch et al. (2020). Observations are provided as initial and goal states ($64 \times 64$ images) and a reward $0 < R \le 1$ upon reaching the goal position. Each episode lasts 400 steps before terminating with a 0 reward. We use the same evaluation parameters as the previous benchmarks and average across 100 runs Pertsch et al. (2020). We also test our method in momentum-based environments, such as RoboYoga Mendonca et al. (2021), which can be challenging due to the lack of momentum information in single images (the goal).

**Agent Hyperparameters**

We use a common hyperparameter setup for all tasks. The goal refresh rate is set to $K = 8$, and the modeled temporal resolutions as $Q = \{2K, 4K, 8K, 16K, 32K\}$. The depth of the unrolled tree during training is $D = 5$ and during inference is $D_I = 8$ unless specified otherwise. For the first 3M steps, the explorer is used as the manager; then it shifts to the planning policy. The agent is trained every 16 environmental steps. Please refer to section B for complete training details.

Figure 5: Full maze with a sample run using our agent.

### 3.1 RESULTS

Figure 6 shows the sample solutions generated by our agent during training and inference. Our agent can navigate the maze to reach far goals successfully and is interpretable.

We compare the performance in terms of the average success rate in reaching the goal state and the average path length against previous methods. Goal-Conditioned Behavioral Cloning (**GC BC**, Nair et al. (2017)) that learns goal-reaching behavior from example goal-reaching behavior. Visual foresight (**VF**, Ebert et al. (2018)) that optimizes rollouts from a forward prediction model via the cross-entropic method (CEM, Rubinstein and Kroese (2004); Nagabandi et al. (2020)). Hierarchical

| Agent | Success rate | Average episode length | Compute steps | Time complexity |
|---|---|---|---|---|
| GC BC | 7% | 402.48 | 1 | 1 |
| GC-Director | 9% | $378.89 \pm 87.67$ | 1 | 1 |
| LEXA (Cos) | 20% | $321.04 \pm 153.29$ | 1 | 1 |
| VF | 26% | 362.82 | $MN$ | $N$ |
| GCP | 82% | 158.06 | $(2N+1)M$ | $\log N$ |
| LEXA (Temp) | 90% | $70.34 \pm 111.14$ | 1 | 1 |
| DHP (*Ours*) | 100% | $73.84 \pm 46.54$ | $\log N$ | $\log N$ |

Table 1: Average Performance of different approaches on the 25-room navigation task over 100 evaluation runs. $N$ is the number of plan steps, and $M$ is the number of samples generated per plan.

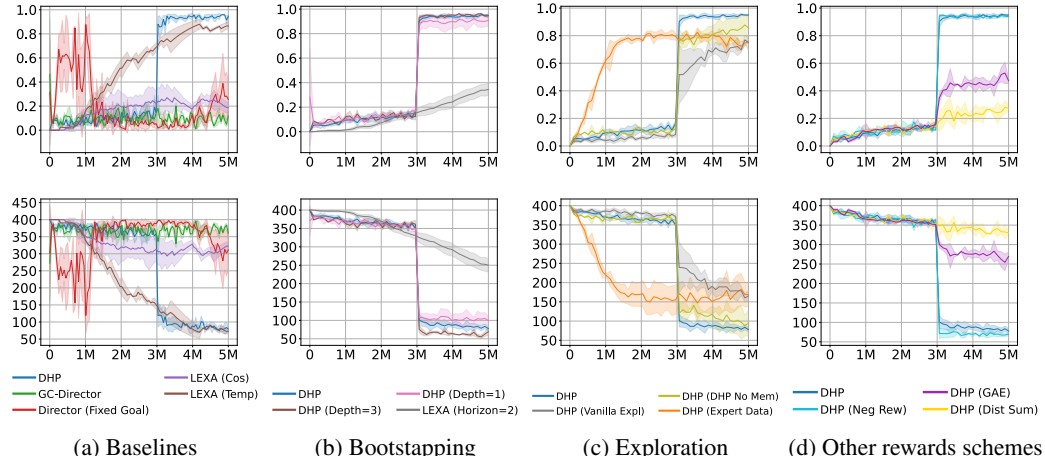

(a) Baselines  (b) Bootstapping  (c) Exploration  (d) Other rewards schemes

Figure 7: Experimental results (mean and stddev across 3 seeds) shown with episodic reward (*top* row) and average episode length (*bottom* row) against the environmental steps. The sharp rise in performance for our methods indicates the switch from exploration to planning. (a) **Baselines:** Comparing our method DHP with state-of-the-art methods. (b) **Bootstrapping:** Compares shallow training depth models ($D = 1, 3$) with the default training depth. (c) **Explore data** Comparison of planning policy performance trained on different data. (d) Comparison to **Other reward schemes**.

planning using Goal-Conditioned Predictors (**GCP**,Pertsch et al. (2020)) optimized using CEM to minimize the predicted distance cost. Goal-Conditioned Director (**GC-Director**). And **LEXA**, a SOTA sequential planning method that also uses an explorer and a planner but optimizes continuous rewards *cosine* and *temporal*. GC BC, VF, and GCP performances are taken from Pertsch et al. (2020), which uses the same evaluation strategy. Table 1 shows that our model outperforms the previous work in terms of success rate and average episode lengths. Our method and **LEXA (Temp)** yield the shortest episode lengths. However, note that while LEXA explicitly optimizes for path lengths, our method uses an implicit objective.

Fig. 7a shows the score and episode length plots for some methods. Here, we plot an extra experiment, **Director (Fixed Goal)**, where the goal remains fixed and the agent only inputs the current state image. It can be seen that the agent shows signs of learning, the score falls but rises again around 8M steps to $\sim 70\%$. Comparing this to **GC-Director** (which completely fails) shows that the issue is not navigation or agent size, but the complexity of a goal-conditioned long-horizon task that requires planning.

## 3.2 ABLATIONS

*Can the planning method generalize to higher depths?*
We train two DHP agents with a maximum tree depth of 3 and 1 during training for planning. The 1 depth agent is only allowed to break the given task once for learning. Fig. 7b shows the comparison of the agents, and it can be seen that all agents perform similarly. Note that LEXA with an equivalent horizon does not perform well.

*Does the complex exploration strategy help?*
We compare the performance of the planning policy trained against three variants: using expert data (collected using a suboptimal policy in Pertsch et al. (2020)), vanilla exploration objective using the reconstruction errors from the unconditional VAE (Sec. 2.4), and using an explorer without the memory. Fig. 7c shows the comparison, and it can be seen that the default agent performs noticeably better.

*Can the method perform in other environments?*
We test our agent in the Deepmind Control Suite Tassa et al. (2018) based RoboYoga Mendonca et al. (2021) environment. The task requires `walker` and `quadruped` agents to reach a goal body orientation specified by an image. The environment is challenging because it rewards the agent for maintaining the goal position, whereas our agent only plans to reach the goal state. Also, the goal states do not encode the momentum information. Fig. 8 shows the performance, of our agent at the tasks. Our agent can solve

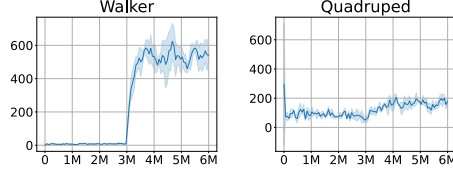

Figure 8: Robo Yoga Environments

the `walker` task, but struggles at `quadruped`. We observed that our `walker` agent maintains the overall pose but constantly sways about the goal position. Fig. 12 shows an episode where the agent headstands constantly. Looking at the scaling limitations of our method, we implement a minimal version of DHP (Sec. F) and evaluate at the OGBench long-horizon large and giant mazes. The resulting agent outperforms the current best methods with an *absolute* improvement of $+71.2\%$ in the giant humanoid maze.

*How does the method perform with other reward and return schemes?*
We compare the default agent against a few variants: **DHP (Neg Rew)** That rewards $-1$ at non-terminal nodes and $0$ at terminal (like an existence penalty), **DHP (GAE)** that estimates advantages as $\min$ of the GAE advantages for each child node, and **DHP (Dist Sum)** rewards negative estimated distances (between $n_{i,0}$ and $n_{i,1}$) at each node. Fig. 7d shows that the **Neg Rew** agent performs as well as the default method, while the others don't perform as well.

## 4 RELATED WORK

**Hierarchical RL Agents:** Hierarchical reinforcement learning (HRL) is a set of techniques that abstracts actions Botvinick et al. (2009); Wiering and Van Otterlo (2012); Barto and Mahadevan (2003); Sutton et al. (1999); Pateria et al. (2021); Nachum et al. (2018). Foundational works, such as the options framework Sutton et al. (1999) and MAXQ decomposition Dietterich (2000), introduced temporal abstraction, enabling agents to reason at multiple time scales. Modern approaches learn hierarchical policies through mutual information (Causal InfoGAN Kurutach et al. (2018), DADS Sharma et al. (2020)), latent space planning (Director Hafner et al. (2022)), or trajectory encoding (OPAL Ajay et al. (2021)). These results demonstrate that hierarchical decomposition facilitates efficient credit assignment in planning.

**Planning Algorithms:** Planning methods aim to solve long-horizon tasks efficiently by exploring future states and selecting optimal actions LaValle (2006); Choset et al. (2005). Monte Carlo Tree Search (MCTS) Browne et al. (2012) expands a tree of possible future states by sampling actions and simulating outcomes. While effective in discrete action spaces, MCTS struggles with scalability in high-dimensional or continuous environments. Visual Foresight methods Ebert et al. (2018); Finn and Levine (2017); Hafner et al. (2019) learned visual dynamics models to simulate future states, enabling planning in pixel-based environments. However, they require accurate world models and can be computationally expensive. Some use explicit graph search over the replay buffer data Eysenbach et al. (2019). Model Predictive Control (MPC) Nagabandi et al. (2018; 2020) is an online planner that samples future trajectories and optimizes actions over a finite horizon. These methods rely on sampling the future state and thus do not scale well with the horizon length. LEXA Kaelbling (1993) is a policy-based linear planner that also uses an explorer for data collection.

To address the challenges of long-horizon tasks, some planning algorithms decompose complex problems into manageable subtasks by predicting intermediate subgoals Parascandolo et al. (2020); Jurgenson et al. (2020); Pertsch et al. (2020). MAXQ Dietterich (2000) decomposes the value function of the target Markov Decision Process (MDP) into an additive combination of smaller MDPs.

Sub-Goal Trees Jurgenson et al. (2020) and CO-PILOT Ao et al. (2021) learn a subgoal prediction policy optimized to minimize the total predicted distance measures of the decomposed subtasks. GCP Pertsch et al. (2020) and kSubS Czechowski et al. (2021) use specialized subgoal predicting modules to search for plans. DHRL Lee et al. (2022) uses efficient graph search by decoupling the time horizons of high-level and low-level policies.

While these methods have demonstrated success, they rely heavily on distance-based metrics, which are challenging to learn and sensitive to policy quality Eysenbach et al. (2019); Ao et al. (2021). In contrast, our method utilizes discrete reachability-based rewards, which are easier to accurately estimate and provide clearer learning signals.

## 5 DISCUSSION & FUTURE WORK

DHP architecture enables us to train a future-conditioned (goal) planning policy, $\pi_P$, and a past-conditioned (memory) exploration policy, $\pi_E$. The resulting model performs expertly on the standard 25-room long-horizon task than the current SOTA approaches (Fig. 7a). The ablations show that the method generalizes beyond the training depths (Fig. 7b), the exploratory rewards significantly impact performance, which are further enhanced using memory (Fig. 7c), and possible alternate reward schemes (Fig. 7d). Our method relies on imagination for training, which allows for plan evaluation without the need to access the internal environmental state; however, it may introduce inaccuracies (Fig. 8). However, we observed better results with discrete rewards than with the distance-based approach for our model (Fig. 7d). The architecture is flexible, and components can be used in isolation, e.g., the planning policy optimization can be combined with custom reachability measures. For future work, the agent can benefit from better and generic goal-state estimation mechanisms, preferably multimodal. The agent can also learn to generate goals for itself via a curriculum for targeted exploration, which can benefit exploration in complex environments with multiple bottleneck states that require a long-term commitment to a goal for efficient exploration. Given this, we demonstrate that the agent can be beneficial in solving long-range tasks and believe that the ideas presented in the paper can be valuable to the community independently. The code for the agent is available at: <URL redacted>.

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

# A   THEORETICAL PROOFS AND DERIVATIONS

## A.1   POLICY GRADIENTS FOR TREES

Given a goal-directed task as a pair of initial and final states $(s_t, s_g)$, a subgoal generation method predicts an intermediate subgoal $s_0$ that breaks the task into two simpler subtasks $(s_t, s_0)$ and $(s_0, s_g)$. The recursive application of the subgoal operator further breaks down the task, leading to a tree of subtasks $\tau$, where each node $n_i$ represents a task. Let the preorder traversal of the subtask tree $\tau$ of depth $D$ be written as $n_0, n_1, n_2, ..., n_{2^{D+1}-2}$. The root node $n_0$ is the given task, and the other nodes are the recursively generated subtasks. Ideally, the leaf nodes should indicate the simplest reduction of the subtask that can be executed sequentially to complete the original task. The tree can be viewed as a trajectory, where each node $n_i$ represents a state, and taking action $\pi_P(z_i|n_i)$ simultaneously places the agent in two states, given by the child nodes $(n_{2i+1}, n_{2i+2})$. Thus, the policy function can be written as $\pi_P(z_i|n_i)$, the transition probabilities (can be deterministic also) as $p_T(n_{2i+1}, n_{2i+2}|z_i, n_i)$, and the probability of the tree trajectory under the policy $\tau_{\pi_P}$ can be represented as:

$$
\begin{aligned}
p_{\pi_P}(\tau) = p(n_0) &* [\pi_P(z_0|n_0) * p_T(n_1, n_2|z_0, n_0)]* \\
&[\pi_P(z_1|n_1) * p_T(n_3, n_4|z_1, n_1)]* \\
&[\pi_P(z_2|n_2) * p_T(n_5, n_6|z_2, n_2)] * ... \\
&[\pi_P(z_{2^D-2}|n_{2^D-2}) * p_T(n_{2^{D+1}-3}, n_{2^{D+1}-2}|z_{2^D-2}|n_{2^D-2})]
\end{aligned}
$$

$$
p_{\pi_P}(\tau) = p(n_0) \prod_{i=0}^{2^D-2} \pi_P(z_i|n_i) \prod_{i=0}^{2^D-2} p_T(n_{2i+1}, n_{2i+2}|z_i, n_i)
$$

**Theorem A.1** (Policy Gradients). *Given a tree trajectory $\tau$ specified as a list of nodes $n_i$, generated using a policy $\pi_P$. The policy gradients can be written as:*

$$
\nabla_{\pi_P} J(\pi_P) = \mathbb{E}_\tau \sum_{i=0}^{2^D-2} A^i(\tau) \nabla_{\pi_P} \log \pi_{\pi_P}(z_i|n_i)
$$

*Proof.* The log-probabilities of the tree trajectory and their gradients can be written as:

$$
\log p_{\pi_P}(\tau) = \log p(n_0) + \sum_{i=0}^{2^D-2} \log \pi_P(z_i|n_i) + \sum_{i=0}^{2^D-2} \log p_T(n_{2i+1}, n_{2i+2}|z_i, n_i) \tag{14}
$$

$$
\nabla_{\pi_P} \log p_{\pi_P}(\tau) = 0 + \nabla_{\pi_P} \sum_{i=0}^{2^D-2} \log \pi_P(z_i|n_i) + 0 = \sum_{i=0}^{2^D-2} \nabla_{\pi_P} \log \pi_P(z_i|n_i) \tag{15}
$$

The objective of policy gradient methods is measured as the expectation of advantage or some scoring function $A(\tau)$:

$$
J(\pi_P) = \mathbb{E}_\tau A(\tau) = \sum_\tau A(\tau) \cdot p_{\pi_P}(\tau) \tag{16}
$$

Then the gradients of the objective function $\nabla_{\pi_P} J(\pi_P)$ wrt the policy $\pi_P$ can be derived as:

$$\nabla_{\pi_P} J(\pi_P) = \nabla_{\pi_P} \sum_{\tau} A(\tau) \cdot p_{\pi_P}(\tau)$$

$$= \sum_{\tau} A(\tau) \cdot \nabla_{\pi_P} p_{\pi_P}(\tau)$$

$$= \sum_{\tau} A(\tau) \cdot p_{\pi_P}(\tau) \frac{\nabla_{\pi_P} p_{\pi_P}(\tau)}{p_{\pi_P}(\tau)}$$

$$= \sum_{\tau} A(\tau) \cdot p_{\pi_P}(\tau) \nabla_{\pi_P} \log p_{\pi_P}(\tau)$$

$$= \mathbb{E}_{\tau} A(\tau) \cdot \nabla_{\pi_P} \log p_{\pi_P}(\tau)$$

$$= \mathbb{E}_{\tau} \sum_{i=0}^{2^D-2} A^i(\tau) \nabla_{\pi_P} \log \pi_P(z_i|n_i) \quad \text{(Using Eq. 15)}$$

$\square$

**Theorem A.2** (Baselines). *If $A(\tau)$ is any function independent of policy actions $z_i$, say $b(n_i)$, then its net contribution to the policy gradient is* 0.

$$\mathbb{E}_{\tau} \sum_{i=0}^{2^D-2} b(n_i) \nabla_{\pi_P} \log \pi_P(z_i|n_i) = 0$$

*Proof.* If $A(\tau)$ is any fixed function that does not depend on the actions $\pi_P(z_i|n_i)$ and only on the state, say $b(n_i)$. Then $b(n_i)$ will be independent of the trajectory $\tau$, and it can be sampled from the steady state distribution under policy $\rho_{\pi_P}$ for any state $n_i$ without knowing $\tau$. In that case,

$$\mathbb{E}_{\tau} \sum_{i=0}^{2^D-2} b(n_i) \nabla_{\pi_P} \log \pi_P(z_i|n_i) = \sum_{i=0}^{2^D-2} \mathbb{E}_{\tau}[b(n_i) \nabla_{\pi_P} \log \pi_P(z_i|n_i)]$$

$$= \sum_{i=0}^{2^D-2} \mathbb{E}_{n_i \sim \rho_{\pi_P}} \mathbb{E}_{a \sim \pi_P}[b(n_i) \nabla_{\pi_P} \log \pi_P(z_i|n_i)]$$

$$= \sum_{i=0}^{2^D-2} \mathbb{E}_{n_i \sim \rho_{\pi_P}}[b(n_i) \mathbb{E}_{a \sim \pi_P} \nabla_{\pi_P} \log \pi_P(z_i|n_i)]$$

$$= \sum_{i=0}^{2^D-2} \mathbb{E}_{n_i \sim \rho_{\pi_P}}[b(n_i) \sum_{a} \pi_P(z_i|n_i) \frac{\nabla_{\pi_P} \pi_P(z_i|n_i)}{\pi_P(z_i|n_i)}]$$

$$= \sum_{i=0}^{2^D-2} \mathbb{E}_{n_i \sim \rho_{\pi_P}} b(n_i)[\nabla_{\pi_P} \sum_{a} \pi_P(z_i|n_i)]$$

$$= \sum_{i=0}^{2^D-2} \mathbb{E}_{n_i \sim \rho_{\pi_P}} b(n_i)[\nabla_{\pi_P} 1] \quad \text{(Sum of probabilities is 1)}$$

$$= 0$$

$\square$

### A.2 POLICY EVALUATION FOR TREES

We present the return and advantage estimation for trees as an extension of current return estimation methods for linear trajectories. As the return estimation for a state $s_t$ in linear trajectories depends upon the next state $s_{t+1}$, our tree return estimation method uses child nodes $(n_{2i+1}, n_{2i+2})$ to compute the return for a node $n_i$. We extend the previous methods, like lambda returns and Gen realized Advantage estimation (GAE) for trees.

The objective of our method is to reach nodes that are directly reachable. Such nodes are marked as terminal, and the agent receives a reward. For generalization, let's say that when the agent takes an action $z_i$ at node $n_i$, it receives a pair of rewards $(R_{2i+1}(n_i, z_i), R_{2i+2}(n_i, z_i))$ corresponding to the child nodes. Formally, the rewards $R(\tau)$ are an array of length equal to the length of the tree trajectory with $R_0 = 0$. Then, the agent's task is to maximize the sum of rewards received in the tree trajectory $\mathbb{E}_\tau \sum_{i=0}^\infty R_i$. To consider future rewards, the returns for a trajectory can be computed as the sum of rewards discounted by their distance from the root node (depth), $\mathbb{E}_\tau \sum_{i=0}^\infty \gamma^{\lfloor \log_2(i+1) \rfloor - 1} R_i$. Thus, the returns for each node can be written as the sum of rewards obtained and the discount-weighted returns thereafter:

$$G_i = (R_{2i+1} + \gamma G_{2i+1}) + (R_{2i+2} + \gamma G_{2i+2}) \tag{17}$$

Although this works theoretically, a flaw causes the agent to collapse to a degenerate local optimum. This can happen if the agent can generate a subgoal very similar to the initial or goal state ($\|s_t, s_{\text{sub}}\| < \epsilon$ or $\|s_g, s_{\text{sub}}\| < \epsilon$). A common theme in reward systems for subgoal trees is to have a high reward when the agent predicts a reachable or temporally close enough subgoal. Thus, if the agent predicts a degenerate subgoal, it receives a reward for one child node, and the initial problem carries forward to the other node.

Therefore, we propose an alternative objective that optimizes for the above objective under the condition that both child subtasks $(n_{2i+1}, n_{2i+2})$ get solved. Instead of estimating the return as the sum of the returns from the child nodes, we can estimate it as the minimum of the child node returns.

$$G_i = \min(R_{2i+1} + \gamma G_{2i+1}, R_{2i+2} + \gamma G_{2i+2}) \tag{18}$$

This formulation causes the agent to optimize the weaker child node first and receive discounted rewards if all subtasks are solved (or have high returns). It can also be noticed that the tree return for a node is essentially the discounted return along the linear trajectory that traces the path with the least return starting at that node. Next, we analyze different return methods in the tree setting and try to prove their convergence.

### A.2.1 LAMBDA RETURNS

TD($\lambda$) returns for linear trajectories are computed as:

$$G_t^\lambda = R_{t+1} + \gamma((1 - \lambda)V(s_{t+1}) + \lambda G_{t+1}^\lambda) \tag{19}$$

We propose, the lambda returns for tree trajectories can be computed as:

$$G_i^\lambda = \min(R_{2i+1} + \gamma((1 - \lambda)V(n_{2i+1}) + \lambda G_{2i+1}^\lambda), R_{2i+2} + \gamma((1 - \lambda)V(s_{2i+2}) + \lambda G_{2i+2}^\lambda)) \tag{20}$$

This essentially translates to the minimum of lambda returns using either of the child nodes as the next state. For theoretical generalization, note that the $\min$ operator in the return estimate is over the next states the agent is placed in. Thus, in the case of *linear* trajectories where there is only one next state, the $\min$ operator vanishes and the equation conveniently reduces to the standard return formulation for *linear* trajectories.

Next, we check if there exists a fixed point that the value function approaches. The return operators can be written as:

$$\mathcal{T}^\lambda V(n_i) = \mathbb{E}_\pi[\min(R_{2i+1} + \gamma((1 - \lambda)V(n_{2i+1}) + \lambda G_{2i+1}),$$
$$R_{2i+2} + \gamma((1 - \lambda)V(n_{2i+2}) + \lambda G_{2i+2}))]$$
$$\mathcal{T}^0 V(n_i) = \mathbb{E}_\pi[\min(R_{2i+1} + \gamma V(n_{2i+1}), R_{2i+2} + \gamma V(n_{2i+2}))]$$
$$\mathcal{T}^1 V(n_i) = \mathbb{E}_\pi[\min(R_{2i+1} + \gamma G_{2i+2}, R_{2i+2} + \gamma G_{2i+2})]$$

**Lemma A.3** (Non-expansive Property of the Minimum Operator). *For any real numbers $a, b, c, d$, the following inequality holds:*

$$|\min(a, b) - \min(c, d)| \leq \max(|a - c|, |b - d|).$$

*Proof.* To prove the statement, we consider the minimum operator for all possible cases of $a$, $b$, $c$, and $d$. Let $\min(a, b)$ and $\min(c, d)$ be the minimum values of their respective pairs.

**Case 1:** $a \leq b$ and $c \leq d$

In this case, $\min(a, b) = a$ and $\min(c, d) = c$. The difference becomes:

$$|\min(a, b) - \min(c, d)| = |a - c|.$$

Since $\max(|a - c|, |b - d|) \geq |a - c|$, the inequality holds.

**Case 2:** $a \leq b$ and $c > d$

Here, $\min(a, b) = a$ and $\min(c, d) = d$. The difference becomes:

$$|\min(a, b) - \min(c, d)| = |a - d|.$$

Since $b \geq a$, $|a - d| \leq |b - d| \leq \max(|a - c|, |b - d|)$, and the inequality holds.

**Case 3:** $a > b$ and $c \leq d$

Here, $\min(a, b) = b$ and $\min(c, d) = c$. The difference becomes:

$$|\min(a, b) - \min(c, d)| = |b - c|.$$

Since $a \geq b$, $|b - c| \leq |a - c| \leq \max(|a - c|, |b - d|)$, and the inequality holds.

**Case 4:** $a > b$ and $c > d$

Symmetrical to Case 1.

**Conclusion:**

In all cases, the inequality

$$|\min(a, b) - \min(c, d)| \leq \max(|a - c|, |b - d|)$$

is satisfied. Therefore, the minimum operator is non-expansive. $\square$

**Theorem A.4** (Contraction property of the return operators). *The Bellman operators $\mathcal{T}$ corresponding to the returns are a $\gamma$-contraction mapping wrt. to $\|\cdot\|_\infty$*

$$\|\mathcal{T}V_1 - \mathcal{T}V_2\|_\infty \leq \gamma\|V_1 - V_2\|_\infty$$

*Proof.* We start with the simpler case, $\mathcal{T}^0$. Let $V_1, V_2$ be two arbitrary value functions. Then the max norm of any two points in the value function post update is:

$$
\begin{aligned}
\|\mathcal{T}^0 V_1 - \mathcal{T}^0 V_2\|_\infty =& \|\mathbb{E}_\pi[\min(R_{2i+1} + \gamma V_1(n_{2i+1}), R_{2i+2} + \gamma V_1(n_{2i+2}))] - \\
& \mathbb{E}_\pi[\min(R_{2i+1} + \gamma V_2(n_{2i+1}), R_{2i+2} + \gamma V_2(n_{2i+2}))]\|_\infty \\
=& \|\mathbb{E}_\pi[\min(R_{2i+1} + \gamma V_1(n_{2i+1}), R_{2i+2} + \gamma V_1(n_{2i+2})) - \\
& \min(R_{2i+1} + \gamma V_2(n_{2i+1}), R_{2i+2} + \gamma V_2(n_{2i+2}))]\|_\infty \\
\leq& \|\min(R_{2i+1} + \gamma V_1(n_{2i+1}), R_{2i+2} + \gamma V_1(n_{2i+2})) - \\
& \min(R_{2i+1} + \gamma V_2(n_{2i+1}), R_{2i+2} + \gamma V_2(n_{2i+2}))\|_\infty \\
\leq& \max(\|\gamma V_1(n_{2i+1}) - \gamma V_2(n_{2i+1})\|_\infty, \|\gamma V_1(n_{2i+2}) - \gamma V_2(n_{2i+2})\|_\infty) \\
\leq& \gamma \max(\|V_1(n_{2i+1}) - V_2(n_{2i+1})\|_\infty, \|V_1(n_{2i+2}) - V_2(n_{2i+2})\|_\infty) \\
\leq& \gamma \max(\|V_1(n_j) - V_2(n_j)\|_\infty, \|V_1(n_k) - V_2(n_k)\|_\infty) \\
\leq& \gamma\|V_1 - V_2\|_\infty \quad (\text{merging } \max \text{ with } \|\cdot\|_\infty)
\end{aligned}
$$

A similar argument can be shown for $\|\mathcal{T}^1 V_1 - \mathcal{T}^1 V_2\|_\infty$ and $\|\mathcal{T}^\lambda V_1 - \mathcal{T}^\lambda V_2\|_\infty$. Using the non-expansive property (Th. A.3) and absorbing the $\max$ operator with $\|\cdot\|_\infty$ leads to the standard form for linear trajectories.

$$
\begin{aligned}
\|\mathcal{T}^\lambda V_1 - \mathcal{T}^\lambda V_2\|_\infty &\leq \|\gamma((1-\lambda)(V_1 - V_2) + \lambda(\mathcal{T}^\lambda V_1 - \mathcal{T}^\lambda V_2))\|_\infty \\
&\leq \gamma(1-\lambda)\|V_1 - V_2\|_\infty + \gamma\lambda\|\mathcal{T}^\lambda V_1 - \mathcal{T}^\lambda V_2\|_\infty \quad \text{(Using triangle inequality)} \\
(1-\gamma\lambda)\|\mathcal{T}^\lambda V_1 - \mathcal{T}^\lambda V_2\|_\infty &\leq \gamma(1-\lambda)\|V_1 - V_2\|_\infty \\
\|\mathcal{T}^\lambda V_1 - \mathcal{T}^\lambda V_2\|_\infty &\leq \frac{\gamma(1-\lambda)}{1-\gamma\lambda}\|V_1 - V_2\|_\infty
\end{aligned}
$$

For contraction, $\frac{\gamma(1-\lambda)}{1-\gamma\lambda} < 1$ must be true.

$$
\begin{aligned}
\frac{\gamma(1-\lambda)}{1-\gamma\lambda} &< 1 \\
\gamma(1-\lambda) &< 1 - \gamma\lambda \\
\gamma - \gamma\lambda &< 1 - \gamma\lambda \\
\gamma &< 1
\end{aligned}
$$

Which is always true.

Since $\mathcal{T}^1$ is a special case of $\mathcal{T}^\lambda$, it is also a contraction. $\qquad\square$

### A.2.2 BOOTSTRAPPING WITH D-DEPTH RETURNS

When the subtask tree branches end as terminal (or are masked as reachable), the agent receives a reward of $1$, which provides a learning signal using the discounted returns. However, when the branches do not end as terminal nodes, it does not provide a learning signal for the nodes above it, as the return is formulated as $\min$ of the returns from child nodes. In this case, we can replace the returns of the non-terminal leaf nodes with their value estimates. Therefore, in the case that the value estimate from the end node is high, indicating that the agent knows how to solve the task from that point onwards, it still provides a learning signal. The $n$-step return for a linear trajectory is written as:

$$
G_t^{(n)} = R_{t+1} + \gamma G_{t+1}^{(n-1)}
$$

with the base case as:

$$
G_t^{(1)} = R_{t+1} + \gamma V(s_{t+1})
$$

We write the $n$-step returns for the tree trajectory as:

$$
G_i^{(d)} = \min(R_{2i+1} + \gamma G_{2i+1}^{(d-1)}, R_{2i+2} + \gamma G_{2i+2}^{(d-1)})
$$

with the base case as:

$$
G_i^{(1)} = \min(R_{t+1} + \gamma V(n_{2i+1}), R_{t+2} + \gamma V(n_{2i+2}))
$$

Value estimates help bootstrap at the maximum depth of the unrolled subtask tree $D$ and allow the policy to learn from incomplete plans.

### A.2.3 PROPERTIES OF TREE RETURN ESTIMATES

In section A.2 it can be seen how the tree return formulation for a node essentially reduces to the linear trajectory returns along the path of minimum return in the subtree under it. When the value function has reached the stationary point. For a subtask tree, if all branches end as terminal nodes, the return

will be $\gamma^{D'}1$, where $D'$ is the depth of the deepest non-*terminal* node. Otherwise, it would be $\gamma^D V'$ where $V'$ is the non-*terminal truncated* node with the minimum return. Thus, it can be seen how higher depth penalizes the returns received at the root node with the discount factor $\gamma$. This property holds for linear trajectories, where the policy converges to the shortest paths to rewards, thereby counteracting discounting Sutton and Barto (2018); Puterman (2014). Thus, our goal-conditioned policy similarly converges to plans trees with minimum maximum depth: $\min_{\pi_P}(\max d_i)$, where $d_i$ is the depth of node $n_i$.

This property also implies that the returns for a balanced tree will be higher than those for an unbalanced tree. The same sequence of leaf nodes can be created using different subtask trees. When the policy does not divide the task into roughly equally tough sub-tasks it results in an unbalanced tree. Since the tree is constrained to yield the same sequence of leaf nodes, its maximum depth $D_U$ will be higher than or equal to a balanced tree $D_U \geq D_B$. Thus, at optimality, the policy should subdivide the task in roughly equal chunks. However, it is worth noting that two subtask trees with different numbers of leaf nodes can have the same maximum depth.

# B  ARCHITECTURE & TRAINING DETAILS

## B.1  WORKER

The worker is trained using $K$-step imagined rollouts ($\kappa \sim \pi_W$). Given the imagined trajectory $\kappa$, the rewards for the worker $R_t^W$ are computed as the `cosine_max` similarity measure between the trajectory states $s_t$ and the prescribed worker goal $s_{\mathrm{wg}}$. First, discounted returns $G_t^\lambda$ are computed as $n$-step lambda returns (Eq. 22). Then the Actor policy is trained using the REINFORCE objective (Eq. 23) and the Critic is trained to predict the discounted returns (Eq. 24).

$$R_t^W = \mathrm{cosine\_max}(s_t, s_{\mathrm{wg}}) \tag{21}$$

$$G_t^\lambda = R_{t+1}^W + \gamma_L((1-\lambda)v(s_{t+1}) + \lambda G_{t+1}^\lambda) \tag{22}$$

$$\mathcal{L}(\pi_W) = -\mathbb{E}_{\kappa \sim \pi_W} \sum_{t=0}^{K-1} \left[ (G_t^\lambda - v_W(s_t)) \log \pi_W(z|s_t) + \eta \mathrm{H}[\pi_W(z|s_t)] \right] \tag{23}$$

$$\mathcal{L}(v_W) = \mathbb{E}_{\kappa \sim \pi_W} \sum_{t=0}^{K-1} (v_W(s_t) - G_t^\lambda)^2 \tag{24}$$

## B.2  EXPLORER

The ELBO objective for the unconditional state recall (UCSR) module is given as:

$$\mathcal{L}(\mathrm{Enc}_U, \mathrm{Dec}_U) = \|s_t - \mathrm{Dec}_U(z_t)\|^2 + \beta \mathrm{KL}[\mathrm{Enc}_U(z_t|s_t) \parallel p_U(z)] \quad \text{where} \quad z_t \sim \mathrm{Enc}_U(z_t|s_t) \tag{25}$$

## B.3  GOAL STATE REPRESENTATIONS

Since the RSSM integrates a state representation using a sequence of observations, it does not work well for single observations. To generate goal state representations using single observations, we train an MLP separately that tries to approximate the RSSM outputs ($s_t$) from the single observations ($o_t$). We refer to these representations as static state representations. Moreover, since GCSR modules require state representations at large temporal distances, it can be practically infeasible to generate them using RSSM. Thus, we use static state representations to generate training data for the GCSR module as well. The MLP is a dense network with a `tanh` activation at the final output layer. It is trained to predict the RSSM output (computed using a sequence of images) using single-image observations. To avoid saturating gradients, we use an MSE loss on the preactivation layer using labels transformed as $l_{\mathrm{new}} = \mathrm{atanh}(\mathrm{clip}(l, \delta - 1, 1 - \delta))$. The clipping helps avoid computational overflows; we use $\delta = 10^{-4}$.

| Name | Symbol | Value |
|------|--------|-------|
| Train batch size | $B$ | 16 |
| Replay trajectory length | $L$ | 64 |
| Replay coarse trajectory length | $L_c$ | 48 |
| Worker abstraction length | $K$ | 8 |
| Explorer Imagination Horizon | $H_E$ | 64 |
| Return Lambda | $\lambda$ | 0.95 |
| Return Discount (tree) | $\gamma$ | 0.95 |
| Return Discount (worker & explorer) | $\gamma_L$ | 0.99 |
| State similarity threshold | $\Delta_R$ | 0.7 |
| Plan temporal resolutions | $Q$ | $\{16, 32, 64, 128, 256\}$ |
| Maximum Tree depth during training | $D$ | 5 |
| Maximum Tree depth during inference | $D_{\mathrm{Inf}}$ | 8 |
| Target entropy | $\eta$ | 0.5 |
| KL loss weight | $\beta$ | 1.0 |
| GCSR latent size | - | $4 \times 4$ |
| RSSM deter size | - | 1024 |
| RSSM stoch size | - | $32 \times 32$ |
| Optimizer | - | Adam |
| Learning rate (all) | - | $10^{-4}$ |
| Adam Epsilon | - | $10^{-6}$ |
| Optimizer gradient clipping | - | 1.0 |
| Weight decay (all) | - | $10^{-2}$ |
| Activations | - | LayerNorm + ELU |
| MLP sizes | - | $4 \times 512$ |
| Train every | - | 16 |
| Prallel Envs | - | 1 |

Table 2: Agent Hyperparameters

## B.4 IMPLEMENTATION DETAILS

We implement two functions: `policy` and `train`, using the hyperparameters shown in Table 2. The agent is implemented in Python/Tensorflow with XLA JIT compilation. Using XLA optimizations, the total training wall time is $2 - 3$ days on a consumer GPU (NVIDIA 4090 RTX 24gb). During inference, the compiled policy function runs in $4.6$ms on average, enabling real-time replanning at $217.39$ times per second (assuming minimal environmental overhead).

### B.4.1 POLICY FUNCTION

At each step, the policy function is triggered with the environmental observation $o_t$. The RSSM module processes the observation $o_t$ and the previous state $s_{t-1}$ to yield a state representation $s_t$. During exploration, the manager $\pi_E$ uses the $s_t$ to generate a worker goal using the unconditional VAE. During task policy, the planning manager $\pi_P$ generates subgoals in the context of a long-term goal $s_g$, and the first directly reachable subgoal is used as the worker's goal. Finally, the worker generates a low-level environmental action using the current state and the worker goal $(s_t, s_{\mathrm{wg}})$. The algorithm is illustrated in (Alg. 1)

### B.4.2 TRAIN FUNCTION

The training function is executed every 16-th step. A batch size $B$ of trajectories $\kappa$ and coarse trajectories $\kappa_c$ is sampled from the exploration trajectories or the expert data. The length of extracted trajectories is $L$ and the length of coarse trajectories is $L_c$ spanning over $L_c \times K$ time steps. Then the individual modules are trained sequentially (Alg. 6):

- RSSM module is trained using $\kappa$ via the original optimization objective Hafner et al. (2019) followed by the static state representations (Sec. B.3).

- The GCSR module is trained using the coarse trajectory $\kappa_c$ (Sec. 2.2).

---

**Algorithm 1** DHP Policy Function

---

**Require:** Observation $o_t$, Goal $o_g$, agent state $t$, $s_{t-1}$, $a_{t-1}$, $s_{wg}$, mem_buff, mode
**Ensure:** Action $a_t$, new agent state

1:  $s_t \leftarrow \text{wm}(o_t, s_{t-1}, a_{t-1})$                                                    {World model update}
2:  **if** $t \mod K = 0$ **then**
3:     **if** mode $== eval$ **then**
4:        $s_g \leftarrow \text{static\_state}(o_g)$                                           {Sec B.3}
5:        $d \leftarrow 0$                                                         {Subgoal planning}
6:        **while** $\neg\text{is\_reachable}(s_t, s_g) \wedge d < D_I$ **do**
7:           $z \sim \pi_P(z|s_t, s_g)$
8:           $s_g \leftarrow \text{Dec}_G(z, s_t, s_g)$
9:           $d \leftarrow d + 1$
10:      **end while**
11:      $s_{wg} \leftarrow s_g$
12:    **else**
13:       $\text{mem}_t \leftarrow \text{extract\_mem}(\text{mem\_buff})$                    {Extract memory Sec. 2.4.2}
14:       $z \sim \pi_E(z|s_t, \text{mem}_t)$
15:       $s_{wg} \leftarrow \text{Dec}_U(z)$
16:    **end if**
17:    $\text{mem\_buff} \leftarrow \text{concat}(\text{mem\_buff}[1:], s_t)$
18:  **end if**
19:  $a_t \sim \pi_W(a_t|s_t, s_{wg})$
20:  **return** $a_t, t+1, s_t, a_t, s_{wg}, \text{mem\_buff}$

---

- The worker policy is optimized by extracting tuples $(s_t, s_{t+K})$ from the trajectories $\kappa$ and running the worker instantiated at $s_t$ with worker goal as $s_{t+K}$ (Sec. B.1).

- The planning policy is trained using sample problems extracted as pairs of initial and final states $(s_t, s_g)$ at randomly mixed lengths from $\kappa_c$. Then the solution trees are unrolled and optimized as in Sec. 2.3.

- Lastly, the exploratory policy is also optimized using each state in $\kappa$ as the starting state (Sec. 2.4.3).

---

**Algorithm 2** Training the GCSR Module

---

**Require:** Experience dataset $data$

1:  Initialize triplets $\leftarrow \emptyset$
2:  **for** all window sizes $q \in Q$ **do**
3:    Append extract_triplets($data, q$) to triplets                {Extract $(s_t, s_{t+q/2}, s_{t+q})$}
4:  **end for**
5:  update_gcsr(triplets)                   {Update CVAE using ELBO objective (Eq. 1)}

---

**Algorithm 3** Training the Worker Policy

---

**Require:** Experience dataset $data$

1:  (init, wk_goal) $\leftarrow$ extract_pairs($data, K$)              {Extract $(s_t, s_{t+K})$ state pairs}
2:  traj $\leftarrow$ imagine(init, $K$)                      {On-policy trajectory rollout}
3:  rew $\leftarrow$ cosine_max(traj, wk_goal)                   {Goal similarity reward}
4:  ret $\leftarrow$ lambda_return(rew)                   {Compute $\lambda$-returns (Eq. 22)}
5:  update_worker(traj, ret)        {Update Worker SAC with REINFORCE (Eqs. 23, 24)}

---

---

**Algorithm 4** Training the Planner Policy

---

**Require:** Experience dataset $data$
1: task ← sample_task_pairs($data$)  {Sample $(s_{\text{init}}, s_{\text{goal}})$ at mixed temporal distances}
2: tree ← imagine_plan(task)  {Generate subtask tree (Sec. 2.3.1)}
3: rew, is_term ← is_reachable(tree)  {Node rewards & terminals (Eqs. 2, 3)}
4: ret ← tree_lambda_return(rew, is_term)  {Lambda return for trees (Eq. 6)}
5: update_planner(tree, ret)  {Update Planner SAC with REINFORCE (Eqs. 8, 9)}

---

**Algorithm 5** Training the Explorer Policy

---

**Require:** Experience dataset $data$
1: (init) ← extract_states($data$)  {Extract $s_t$ states as initial states}
2: traj ← imagine(init, $H_E$)  {On-policy trajectory rollout}
3: rew ← expl_rew(traj)  {Exploratory reward (Eq. 10)}
4: ret ← lambda_return(rew)  {Compute $\lambda$-returns (Eq. 11)}
5: update_explorer(traj, ret)  {Update Explorer SAC (Eqs. 12, 13)}

---

**Algorithm 6** Overall Training Procedure

---

**Require:** Experience dataset $data$, current mode $mode$
1: train_rssm($data$)  {Train world model per Hafner et al. (2019)}
2: train_gcsr($data$)  {Alg. 2}
3: train_worker($data$)  {Alg. 3}
4: train_planner($data$)  {Alg. 4}
5: train_explorer($data$)  {Alg. 5}

---

## C  BROADER IMPACTS

### C.1  POSITIVE IMPACTS

The imagination-based policy optimization mitigates hazards that can occur during learning. Efficient training can reduce the carbon footprint of the agents. The agent produces highly interpretable plans that can be verified before execution.

### C.2  NEGATIVE IMPACTS AND MITIGATIONS

- **Inaccurate Training**: Imagination can cause incorrect learning. Mitigation: Rigorous testing using manual verification of world-model reconstructions against ground truths.

- **Malicious Use**: Hierarchical control could enable more autonomous adversarial agents. Mitigation: Advocate for gated release of policy checkpoints.

### C.3  LIMITATIONS OF SCOPE

Our experiments focus on simulated tasks without human interaction. Real-world impacts require further study of reward alignment and failure modes.

# D  SAMPLE EXPLORATION TRAJECTORIES

We compare sample trajectories generated by various reward schemes in this section. Fig. 9 shows the sample trajectories of an agent trained to optimize the vanilla exploratory rewards. Fig. 10 shows the sample exploration trajectories of an agent that optimizes only for the GCSR-based rewards. Fig. 11 shows sample exploration trajectories of an agent that optimizes for GCSR exploratory rewards but without memory. The GCSR rewards-based agent generates trajectories that are less likely to lead to repeated path segments or remaining stationary. Removing the memory can sometimes cause inefficient trajectories, where the agent may not perform as expected.

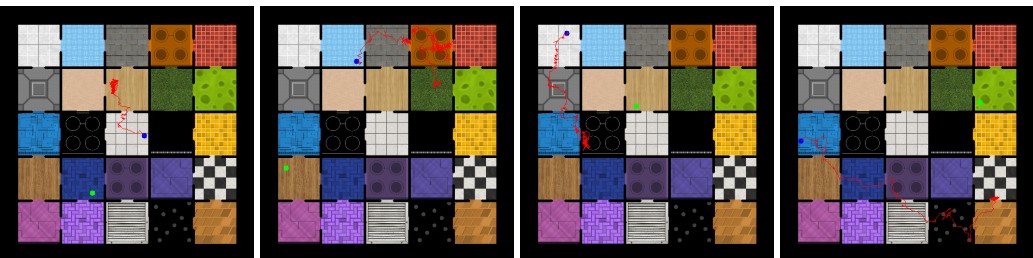

Figure 9: Sample exploration trajectories using the vanilla exploratory rewards. The agent identifies states with unclear representations and successfully navigates to them, indicating sufficient navigation capabilities. However, once the agent reaches the goal, it stays there, leading to lower-quality data.

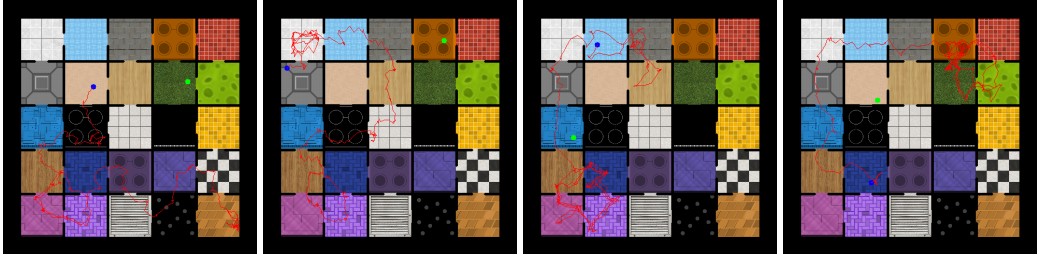

Figure 10: Sample exploration trajectories using the GCSR modules for the path-segments-based rewards. It can be seen that the agent continuously moves and explores longer state connectivity.

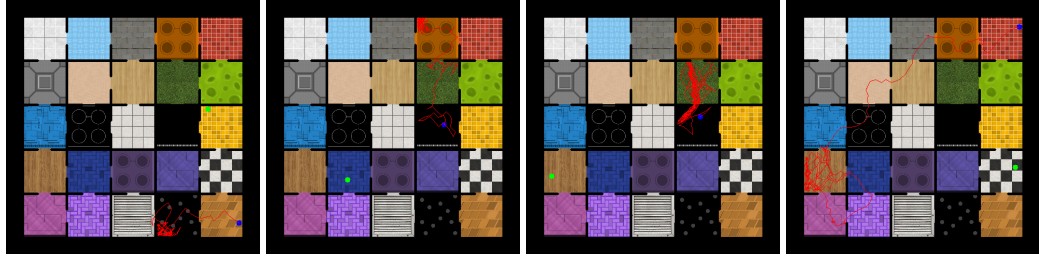

Figure 11: Sample exploration trajectories using the default strategy without the Memory. The resulting data can have some problematic trajectories.

# E  ROBO YOGA SAMPLES

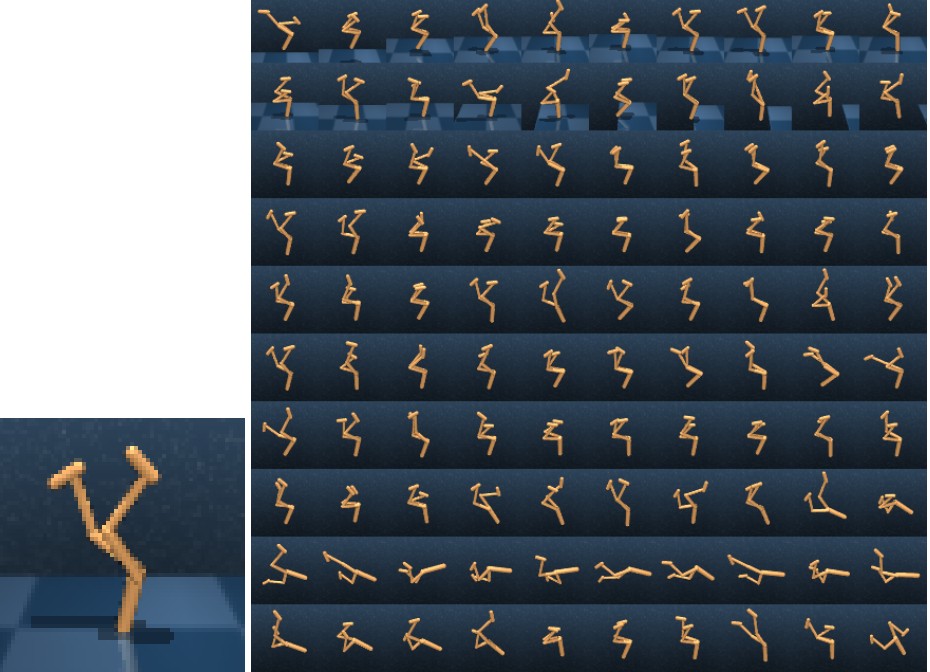

Figure 12: Figure shows a sample trajectory (every 8-th frame) for the `walker` embodiment given a headstand goal. The agent can maintain a constant headstand; however, it sways about the goal position. This is because the agent is trained to reach the goal position and not to stay there.

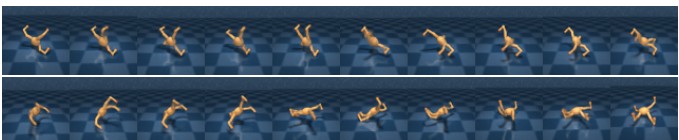

Figure 13: Some failure cases from the plans constructed for the yoga `quadruped` task where the agent hallucinates wrong midway states that get verified by the similarity check. We believe this is because the quadruped agent has a richer state space than the walker, and the states are not very distinct.

# F  OFFLINE DHP

While the online formulation presented in earlier sections is highly flexible and requires minimal external dependencies, scaling to extremely long-horizon and complex tasks (such as OGBench (Park et al., 2024)) reveals practical challenges:

- **Compounding errors in subgoal prediction**: Since subgoals at lower tree levels depend on predictions at higher levels, any hallucinations in the CVAE-based subgoal generation accumulate with depth (Figure 13 shows examples). This becomes problematic when planning over very long horizons where recursive errors compound.

- **Long-horizon exploration inefficiency**: Undirected exploration can require impractically long training times in large environments. Moreover, many states that yield high exploratory rewards may be irrelevant to the external task (e.g., an agent lying upside-down throughout a maze provides novel states but unhelpful coverage).

To address these limitations while preserving our core contributions—discrete reachability checks and tree-structured return estimation—we develop a simplified offline variant of DHP. Demonstrating that our planning principles are architecture-agnostic and applicable across different settings.

## F.1  ARCHITECTURE

The offline agent maintains our core planning objectives while replacing components that proved problematic in complex domains:

1. **Goal representation network** $\phi(s_t, s_g) \to z$ (similar to HIQL (Park et al., 2023)): Replaces world-model-based state representations with a learned encoder that directly maps state-goal pairs to a length-normalized representation space.

2. **Hierarchical policies**:
   - **Manager** $\pi_M(s_t, s_g) \to z$: Predicts optimal subgoal representations in the learned $z$-space.
   - **Worker** $\pi_W(s_t, z) \to a_t$: Executes low-level actions to reach subgoals.

3. **Goal buffer** $\mathcal{B}$. Stores a diverse set of landmark states from the offline dataset. During planning, the buffer retrieves the nearest stored state $\mathcal{B}(s_t, z) \to s_g$, to the predicted representation $z$, acting as a discrete decoder. This eliminates hallucination errors by constraining subgoals to states that actually exist in the data.

## F.2  TRAINING OBJECTIVES

The agent is trained using offline datasets collected by noisy expert policies (Park et al., 2024). All modules are trained jointly using samples from the replay buffer.

**Manager Value Function**  The manager value function $V^h(s_t, s_g)$ evaluates the quality of high-level plans. Following our tree-structured return formulation (Sec. 2.3.3), we compute values using the min-operator.

For a given task $(s_t, s_g)$ decomposed via a midway state $s_w$:

$$V^h(s_t, s_g) = \min(R_{\text{left}} + \gamma^h V^h(s_t, s_w), R_{\text{right}} + \gamma^h V^h(s_w, s_g))$$

where $R_{\text{left}}, R_{\text{right}}$ are binary rewards $\{0, 1\}$ indicating whether each subtask is reachable within the threshold of subgoal step (replacing the imagination-based check with a distance threshold).

The value function is trained using an expectile regression objective (similar to IQL (Kostrikov et al., 2021), HIQL Park et al. (2023)) to avoid overestimation. However, unlike standard IQL, which regresses toward Bellman backup values, we regress toward our tree-structured return $G^h$:

$$G^h(s_t, s_g; s_w) = \min(R_{\text{left}} + \gamma^h \bar{V}^h(s_t, s_w), R_{\text{right}} + \gamma^h \bar{V}^h(s_w, s_g))$$

This is the 1-step tree return from Equation 5, where we use target networks $\bar{V}^h$ for stability. The loss is then:

$$L_{V^h} = \mathbb{E}(s_t, s_w, s_g)[\mathcal{L}\tau(G^h(s_t, s_g; s_w) - V^h(s_t, s_g))]$$

where $\mathcal{L}_\tau$ is the asymmetric expectile loss:

$$\mathcal{L}_\tau(u) = |\tau - 1[u < 0]| \cdot u^2$$

with expectile $\tau = 0.7$. This asymmetric loss causes the value function to approximate an upper expectile of the tree-structured return distribution, avoiding overestimation while maintaining optimism for planning.

**Manager Actor**    The manager policy learns to predict subgoal representations that maximize the advantage:

$$\pi_M(z|s_t, s_g) \propto \exp(\beta \cdot A^h(s_t, s_w, s_g))$$

where $s_w$ is the midway state between $(s_t, s_g)$ in the dataset trajectory, and the advantage is:

$$A^h(s_t, s_w, s_g) = \min(V^h(s_t, s_w), V^h(s_w, s_g)) - V^h(s_t, s_g)$$

This encourages the manager to predict subgoals that create balanced, high-value decompositions. The policy is trained via advantage-weighted regression (AWR):

$$\mathcal{L}_{\pi_M} = -\mathbb{E}_{(s_t, s_w, s_g)}[\exp(\beta \cdot A^h) \cdot \log \pi_M(\phi(s_t, s_w)|s_t, s_g)]$$

where $\beta = 3.0$ is the temperature parameter controlling the strength of advantage weighting.

**Worker Value Function**    The worker value function $V^l(s_t, z)$ where $z = \phi(s_t, s_g)$ evaluates how well the agent can reach subgoal $s_g$ from state $s_t$. It is trained using standard IQL expectile regression:

$$\mathcal{L}_{V^l} = \mathbb{E}_{(s_g, s', s_g)}[\mathcal{L}_\tau(r + \gamma^l V^l(s_t, z) - V^l(s_t, z))]$$

where $r = \text{Float}[d(s', s_g) \leq 1]$ is a sparse binary reward indicating goal achievement, and $\gamma^l = 0.99$ is the discount factor. Similar to HIQL (Park et al., 2023), the goal representation module $\phi$ is trained using the gradients from the value function loss.

**Worker Actor**    The worker policy is trained to imitate actions that have high advantages:

$$L_{\pi_W} = -\mathbb{E}(s, a, s', s_{t+K})[\exp(\beta \cdot (V^l(s', z) - V^l(s_t, z))) \cdot \log \pi_W(a_t|s_t, z)]$$

where $z = \phi(s_t, s_{t+K})$. This encourages the worker to execute actions that make progress toward the subgoal $s_{t+K}$.

**Goal Buffer Update**    The goal buffer is populated and maintained using farthest point sampling (FPS) to ensure diversity:

1. **Initialization**: Sample $N$ candidate goals from the offline dataset
2. **Diversity selection**: Every 10000 steps, we iteratively select states that maximize minimum distance to already-selected states: $s_{\text{next}} = \arg\max_{s \in \text{candidates}} \min_{s' \in \mathcal{B}} d(s, s')$ where $d$ is a value-based distance estimate $-V^l(s, s')$.

The buffer capacity is set to 2048 goals, which we found to work well for all tasks. But it can be task-dependent. Using value-based FPS ensures the buffer contains goals at varying difficulty levels, not just perceptually diverse states.

| Task | GCBC | GCIVL | GCIQL | QRL | CRL | HIQL | DHP *(Ours)* |
|------|------|-------|-------|-----|-----|------|--------------|
| antmaze-large-navigate-v0 | $24 \pm 2$ | $16 \pm 5$ | $34 \pm 4$ | $75 \pm 6$ | $83 \pm 4$ | $91 \pm 2$ | $\mathbf{93.9 \pm 1}$ |
| antmaze-giant-navigate-v0 | $0 \pm 0$ | $0 \pm 0$ | $0 \pm 0$ | $14 \pm 3$ | $16 \pm 3$ | $65 \pm 5$ | $\mathbf{72.3 \pm 5}$ |
| humanoidmaze-large-navigate-v0 | $1 \pm 0$ | $2 \pm 1$ | $2 \pm 1$ | $5 \pm 1$ | $24 \pm 4$ | $49 \pm 4$ | $\mathbf{82.8 \pm 5}$ |
| humanoidmaze-giant-navigate-v0 | $0 \pm 0$ | $0 \pm 0$ | $0 \pm 0$ | $1 \pm 0$ | $3 \pm 2$ | $12 \pm 4$ | $\mathbf{83.2 \pm 4}$ |

Table 3: Performance on OGBench navigation tasks.

### F.3 INFERENCE

During inference, the manager recursively decomposes tasks up to depth $D = 8$. For each decomposition, the manager predicts a goal representation $\pi_M(s_t, s_g) \rightarrow z$. Then, the $z$ is used to retrieve the nearest state by comparing the goal representations of the stored states $\phi(s_t, s_i) \forall s_i \in \mathcal{B}$ with the predicted $z$ using the mean-squared error. The first reachable subgoal in the decomposed subgoal stack (say $s_w$) is used as the worker's subgoal to predict the action as: $\pi_W(s_t, \phi(s_t, s_w))$. Reachability is determined by the normalized value criterion $\tilde{V}^l(s_t, z) > \theta$, where $\tilde{V}^l = (V^l - \mu)/\sigma$ is the value estimate normalized by running statistics $(\mu, \sigma)$ collected during training for valid subgoals $(s_{t+K})$. We set $\theta = -2.0$, which corresponds to subgoals that are in the top $97.5\%$ of value estimates, preventing over-optimism.

### F.4 EVALUATION

We evaluate the offline DHP agent on the OGBench navigation benchmark (Park et al., 2024), which features significantly larger and more complex environments than the 25-room task:

- **AntMaze-Large/Giant:** Large continuous mazes requiring precise locomotion control.
- **HumanoidMaze-Large/Giant:** High-dimensional humanoid morphology (17 DoF) in large mazes, requiring both locomotion and balance.

Table 3 shows that offline DHP achieves state-of-the-art performance, with particularly strong *absolute* gains on the HumanoidMaze tasks ($+33.8\%$ on large, $+71.2\%$ on giant compared to HIQL). Our planning method is highly interpretable, as shown in the subgoal visualization in Fig. 14. For video results, please visit `https://sites.google.com/view/dhp-video/home`. These are the best results on the OGBench humanoid tasks to the best of our knowledge. These results demonstrate that:

- **Core contributions generalize:** The discrete reachability paradigm and tree-structured returns transfer successfully to the offline setting.
- **Architecture modularity:** Our planning principles work with both online (CVAE+world model) and offline (goal buffer+representation learning) implementations.
- **Scalability:** The method handles high-dimensional morphologies and giant environments where the online version struggled.

### F.5 COMPARISON TO ONLINE DHP

The offline variant trades exploration flexibility for robustness:

**Advantages**:

- Eliminates subgoal hallucinations via buffer-based retrieval
- Scales to high-dimensional state spaces (humanoid)
- No world model training required

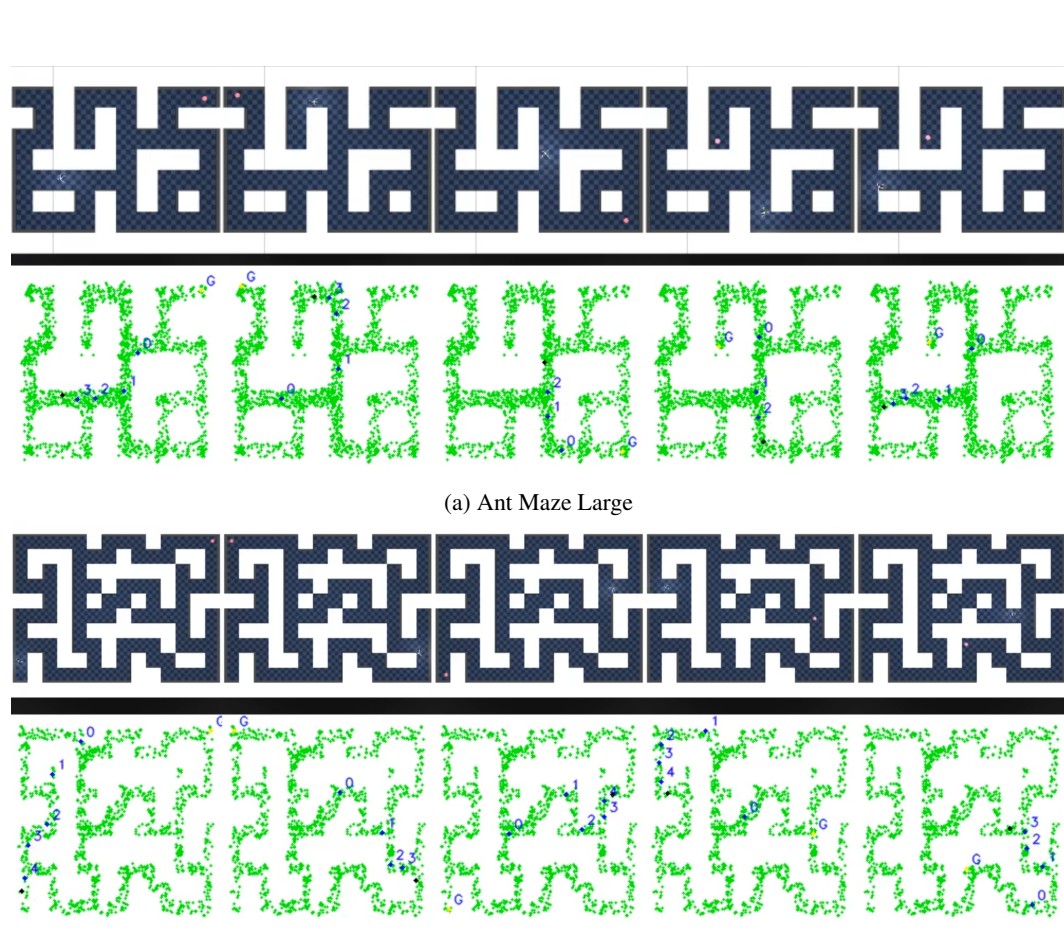

(a) Ant Maze Large

(b) Ant Maze Giant

Figure 14: Samples of plan visualizations at the OGBench navigation tasks. The green points signify the stored states in the buffer $\mathcal{B}$. The black dot specifies the agent's current position. The yellow dot labeled 'G' is the final goal. And the subgoals are represented by blue dots, with their indices in the subgoal stack shown next to each. The agent recursively plans until a reachable subgoal is found, then moves towards it.

- Handles extremely long horizons (giant mazes)

**Tradeoffs**:

- Requires pre-collected offline data
- Subgoal space limited to buffer contents (though 2048 landmarks prove sufficient)
- Cannot discover entirely novel subgoals beyond training distribution

Both variants validate that our core insight—replacing distance metrics with discrete reachability and using tree-structured advantages—is the key to effective hierarchical planning, independent of the specific architecture used.

| $L$ | State space size | Success rate | Path lengths |
|---|---|---|---|
| 2 | 16 | 100% | $2.42 \pm 0.56$ |
| 3 | 512 | 86.47% | $3.12 \pm 1.66$ |

Table 4: Performance at the lights-out puzzle.

## G  LIGHTSOUT PUZZLE

Lightsout is a complex puzzle where the agent is given a binary 2D array representing the state of lights on/off in a grid of rooms. The objective is to turn off the lights in all rooms; but, toggling the lights in room $(i, j)$ toggles the lights of connected rooms $[(i - 1, j), (i + 1, j), (i, j - 1), (i, j + 1)]$. We measure reachability by iterating over all actions to check if the goal state can be reached in one step (errorless check). The agent is required to plan a path from the initial state to the final all-out state.

To assess the planning capability of DHP, the agent is stripped down to just the planning actor and critic, both of which are implemented as simple dense MLPs. The actor outputs directly in the state space. For a grid $L \times L$, the size of the state space is $2^{L^2}$. Thus, the agent must choose the right subgoal state of the possible $2^{L^2}$ at each step in tree planning. This is more complex than the 25-room task, where the agent had to essentially pick from 25 rooms at each step. The agent plans up to a depth of $D = 5$, and all branches must terminate in success (within a depth of $D = 5$) in a single attempt to score $1$; otherwise, it scores $0$. Another added complication is that some subgoals can be dead ends and may not lead to the goal. We observe that our agent performs decently (path lengths refer to the lengths of the successful paths). Table 4 shows the final results.

