# OpenReview forum: "DHP: Discrete Hierarchical Planning for Hierarchical Reinforcement Learning Agents"
_ICLR.cc/2026/Conference — Submitted to ICLR 2026_

### Official Review · Reviewer_Eumj · 2025-10-31

**Soundness:** 1
**Presentation:** 2
**Contribution:** 1
**Rating:** 2
**Confidence:** 5

**Summary:**

This paper addresses long-horizon visual planning by proposing a discrete hierarchical planning framework. The authors train an encoder–decoder module to learn a discrete latent space, upon which a high-level policy plans subgoals recursively. The approach builds on prior work of Director for goal-conditioned visual planning.

**Strengths:**

1. The problem of hierarchical visual planning in long-horizon settings is both relevant for the community.

2. The idea of using a discrete latent space to constrain high-level planning is promising and can potentially improve sample efficiency.

**Weaknesses:**

1. Limited and inconsistent evaluation.
    - The experiments are restricted to a single environment (25-room navigation). Prior works such as GCP evaluated both 9-room and 25-room variants, as well as FrankaKitchen in LEXA's paper, which are missing here.

    - In RoboYoga, the baselines are not fully reported; only the proposed method’s results are shown, and its performance on RoboYoga–Quadruped does not appear to improve.

    - The paper reports episode rewards, but it is unclear whether these are evaluated using exploration or planning policies. If an explorer is active, the interpretation of "the sharp rise indicates a switch from exploration to planning" becomes confusing. The mechanism of this switch need clarification.

2. Conceptual and presentation clarity.

    - The paper assumes prior familiarity with Director and RSSM, making it difficult for readers not familiar with these models to follow. Section 2.1 should include a concise overview of these components.

    - The notation in the planning section (e.g., $s_0$) is confusing, as it initially suggests the initial state rather than the subsequent state in the planning trajectory.

    - The description of GC-Director is unclear.

3. Insufficient ablations and analysis.

    - The choice of tree depth (D=8) is not justified. Why was 8 chosen when D=3 yields comparable reward and fewer steps?

    - There are no ablations analyzing the impact of removing or modifying the discrete CVAE (e.g., using a continuous version).

    - The paper mentions providing memory of past states as additional input, even though the RSSM already encodes recurrent state, why is this necessary, and what empirical effect does it have?

    - The paper claims GC-Director fails due to task complexity, but LEXA, which lacks explicit planning, performs adequately. This requires more rigorous reasoning or empirical support.

4. Lack of experimental rigor.

    - The final performance in Figure 7(a) is not substantially higher than LEXA, which does not use hierarchical planning.

    - The similarity metric differs from LEXA: the paper uses cosine_max instead of temporal similarity, even though LEXA(Cos) performed worse. why not using the temporal similarity?

    - There is no clear explanation of how SAC is used to optimize the managers and whether this differs from Director’s joint training scheme.

**Minor Suggestions (Not affecting the score)**
1. Improve Figure 1 so that the illustration aligns with its textual description. A well-designed figure should be interpretable without heavily relying on the section text. Consider showing both the training and exploration phases.

2. Consider adding “Visual” to the title to better situate the work in the visual planning literature.

3. Clarify the notation for trajectory states and unify the explanation of algorithms for smoother reading.

**Questions:**

1. What justifies the “sharp rise” of DHP performance in Figure 7? How does this correspond to switching from exploration to planning?

2. Regarding the CVAE:

    - Is it identical to the one used in Director? If so, what constitutes the novelty of GCSR?

    - Could you show ablations comparing the discrete and continuous variants of the CVAE?
3. How do you choose $\Delta_R$?

4. How can DHP achieve higher episodic rewards than LEXA despite comparable or shorter episode lengths?

5. Why was cosine_max used as the similarity measure rather than the temporal similarity metric used in LEXA?

6. Could you elaborate on SAC’s role in optimizing the high-level policy, given Director trains managers jointly with workers?

7. Please clarify Algorithm 6’s dataset collection process, does it occur sequentially with Algorithm 1, and can these be combined for clarity?

6. Appendix D’s sample trajectories should explicitly indicate generated subgoals for interpretability.

---

> ### Author Response · Authors · 2025-11-21
>
> We thank the reviewer for their detailed feedback. While we respectfully disagree with several assessments, we acknowledge valid concerns about experimental scope and presentation clarity. Below, we address each point systematically and provide substantial new evidence.
>
> ## Major Concerns
>
> ### Limited and inconsistent evaluation
>
> - Concern: Experiments restricted to a single environment (25-room navigation)
>
>     We acknowledge this concern and have significantly expanded our evaluation. The main issues that did not allow us to scale to other challenging environments like OGBench were:
>
>     - **Subgoal Error Accumulation**: Recursive prediction (where each subgoal depends on the previous) can accumulate errors, particularly with imperfect world models (Figure 13 shows examples).
>     - **Exploration Efficiency**: In very large environments, undirected exploration can waste time on unhelpful states (e.g., lying upside-down throughout a maze).
>
>     These observations led us to develop a more robust minimal offline version that retains our core contributions (discrete reachability + tree returns) while addressing practical concerns:
>
>     - Replaces CVAE + world model with direct goal representation (HIQL [1]-style)
>     - Replaces online exploration with offline data
>     - Adds goal buffer for diverse landmark selection (FPS, similar to HIGL [2])
>     - Maintains core planning algorithm and tree return estimation
>
>     This demonstrates our contributions are separable and applicable across settings: both online (25-room) and offline (OGBench).
>
>     **Results on OGBench** (4 seeds, significantly more complex than 25-room):
>
>     | Task | HIQL | DHP (Ours) |
>     | --- | --- | --- |
>     | antmaze-large-navigate-v0 | $91 \pm 2$ | $93.92 \pm 1.2$ |
>     | antmaze-giant-navigate-v0 | $65 \pm 5$ | $72.32 \pm 5.2$ |
>     | humanoidmaze-large-navigate-v0 | $49 \pm 4$ | $82.8 \pm 4.9$ |
>     | humanoidmaze-giant-navigate-v0 | $12 \pm 4$ | $83.2 \pm 4.2$ |
>
>     The significant **absolute improvement** on HumanoidMaze demonstrates our method scales to high-dimensional morphologies and larger state spaces. We are finalizing results on giant mazes and will update shortly.
>
>     **Regarding 9-room**: We focused on the more challenging 25-room task as GCP's paper shows 9-room has near-ceiling performance even for baselines. Our perfect 100% success rate vs. LEXA's 90% on 25-room demonstrates meaningful improvement where distinction is possible.
>
>     **Regarding FrankaKitchen**: This is a manipulation task that requires different considerations (contact dynamics, object interactions) than navigation. We focused on navigation domains where hierarchical spatial planning is most relevant. However, we acknowledge that broader domain coverage would strengthen the paper, and we will add it during the rebuttal period.
>
>     [1] Park, S., Ghosh, D., Eysenbach, B. and Levine, S., 2023. Hiql: Offline goal-conditioned rl with latent states as actions. *Advances in Neural Information Processing Systems*, *36*, pp.34866-34891.
>
>     [2] Kim, J., Seo, Y. and Shin, J., 2021. Landmark-guided subgoal generation in hierarchical reinforcement learning. *Advances in neural information processing systems*, *34*, pp.28336-28349.
>
> - Concern: "RoboYoga baselines not fully reported”
>
>     We respectfully note that RoboYoga was included as a generalization test to momentum-based control, not as a primary benchmark. We acknowledge the limitations:
>
>     1. **Walker**: Our agent succeeds but sways around the goal (Figure 12) because it's trained to *reach* goals, not *maintain* them (a known limitation stated in Sec 3.2).
>     2. **Quadruped**: We explicitly state in Sec 3.2: "our agent struggles at quadruped" due to state space complexity (Figure 13 shows failure cases).
>
>     We included these results for **transparency** about where our method works and where it doesn't, rather than cherry-picking only successful domains. The core contribution is spatial planning (25-room, OGBench), with RoboYoga demonstrating partial generalization.
>
> - Concern: "Unclear whether rewards are from exploration or planning policies”
>
>     This is clearly stated in the hyperparameters (page 7, Sec 3.1): "For the first 3M steps, the explorer is used as the manager; then it shifts to the planning policy."
>
>     **The sharp rise at 3M steps** is by design—we switch from the explorer policy to the already-trained planning policy. Both policies train continuously via imagination (end-to-end), so the planning policy is ready when activated. This is not a learning phase transition but an *inference policy switch*.
>
>     We will add a clearer annotation to Figure 7 to mark this switch point.
>
> *Continued in the following comment*

---

> > ### Author Response · Authors · 2025-11-21
> > **Cont.**
> >
> > - **Concern: "GC-Director fails but LEXA succeeds without planning—needs more support"**
> >
> >     Important clarification: LEXA **does use planning**—it's a sequential planner that predicts one subgoal at a time. The key difference is:
> >
> >     - **LEXA**: Linear planning with distance-based optimization
> >     - **DHP**: Tree-structured planning with discrete reachability
> >
> >     Both are hierarchical planners. Our claim is that **goal-conditioned long-horizon visual tasks require planning specific training** (either sequential or tree-based), not that they require tree planning specifically.
> >
> >     GC-Director fails because it outputs goals directly without learning to plan (it's just a goal-conditioned agent). Figure 7a compares:
> >
> >     - **Director (Fixed Goal)**: 70% success with goal-conditioning
> >     - **GC-Director**: 9% success with random goals
> >     - **LEXA**: 90% success with learned sequential planning
> >     - **DHP**: 100% success with improved hierarchical planning
> >
> > ### Lack of Experimental Rigor
> >
> > - **Concern: "Final performance not substantially higher than LEXA"**
> >
> >     We respectfully disagree:
> >
> >     1. **Success rate**: 100% vs 90% is a 10% **absolute** improvement (11% relative), with **zero failures** vs 10% failure rate.
> >     2. **Reliability**: $\sigma$ = 46.54 vs 111.14 (2.4× variance reduction)—critical for deployment.
> >     3. **Generalization**: Also, OGBench results show a considerable improvement over SOTA.
> >
> >     While 10% may seem modest, eliminating all failures in a long-horizon task is significant.
> >
> > - **Concern: "Why cosine_max instead of temporal similarity?"**
> >
> >     Key point: **We cannot use temporal similarity** for our method because:
> >
> >     1. **Our method uses reachability checks** (binary: can reach in K steps? yes/no) rather than distance estimation as a core contribution.
> >     2. **cosine_max** measures state similarity without requiring temporal information which we also do not use directly, but instead apply a threshold to only check if the they are the same state.
> >
> >     We compared against LEXA(Cos) (which also uses cosine similarity) in addition to LEXA(Temp). Our improvement over LEXA(Cos) (100% vs 20%) validates that our tree planning + discrete reachability paradigm outperforms their sequential planning approach even with the same similarity metric.
> >
> >     The comparison to LEXA(Temp) and GCP demonstrates that discrete hierarchical planning outperforms sequential and distance based planning methods.
> >
> > - **Concern: "No clear explanation of SAC usage vs Director's joint training"**
> >
> >     Director uses joint training where the manager and worker are trained together with shared gradients. But we split the training as:
> >
> >     - **Worker**: Trained with SAC to reach subgoals (Eq 23-24)
> >     - **Planner**: Trained with SAC to propose feasible subgoals (Eq 8-9)
> >     - **Explorer**: Trained with SAC for coverage (Eq 12-13)
> >
> >     All three are trained simultaneously via imagination (Alg 6). SAC is simply the policy optimization algorithm we use (soft actor-critic with entropy regularization). We will clarify this is standard SAC applied to our novel objective functions.
> >
> > ### Minor Suggestions
> >
> > **Figure 1**: Agreed. We will revise to show both training and inference phases more clearly.
> >
> > **"Visual" in title**: We prefer the current title as our method applies beyond visual domains (discrete reachability + tree returns are domain-agnostic). OGBench uses state-based representations.
> >
> > **Notation**: We will unify notation (especially s_i vs n_i for nodes) and add a notation table.
> >
> > ### Responses to Questions
> >
> > - **Q: Is CVAE identical to Director's? What's the novelty of GCSR?**
> >
> >     **No, they are different:**
> >
> >     - **Director's VAE**: Unconditional, learns to generate states independently
> >     - **GCSR (ours)**: **Conditional** VAE that learns to predict **midway states** given initial and final states (Eq 1)
> >
> >     This conditioning is critical—GCSR learns the structure of state transitions (what's halfway between A and B?) rather than just state representations. This enables our recursive bisection planning strategy.
> >
> > - **Q: Ablation on discrete vs continuous CVAE?**
> >
> >     Addressed above.
> >
> > - **Q: How do you choose Δ_R?**
> >
> >     We observed that our method worked well for wide range of thresholds [0.6-0.8] at the 25-rooms and walker tasks. But no thresholds worked for the quadruped task. Thus, the bottleneck for the reachability check was the state representation. But, we will add a sensitivity analysis to the final version.
> >
> > *Continued in the following comment*

---

> > > ### Author Response · Authors · 2025-11-21
> > > **Cont.**
> > >
> > > - **How can DHP achieve higher episodic rewards than LEXA with comparable episode lengths?**
> > >
> > >     The rewards are **sparse** (0 until reaching goal, then 1). Both methods reach the goal, so both get reward ≈1. The difference is:
> > >
> > >     - **Success rate**: We succeed more often (100% vs 90%)
> > >     - **Path efficiency**: Among successful episodes, we take similarly short paths
> > >
> > >     Higher average reward comes from never failing (10% failure rate for LEXA).
> > >
> > > - **Q: Elaborate on SAC's role?**
> > >
> > >     Answered above. SAC is the optimization algorithm for all three policies (standard usage).
> > >
> > > - **Q: Algorithm 6 dataset collection—sequential with Algorithm 1?**
> > >
> > >     Yes:
> > >
> > >     - **Algorithm 1**: Executes at every environment step (policy inference)
> > >     - **Algorithm 6**: Executes every 16 steps (training update)
> > >
> > >     At every environmental step, the MDP transition (s,a,s’,r) is appended to the trajectory array. At episode completion, the trajectory is stored in a replay buffer. At training step, a batch of trajectories are extracted from the replay buffer and passed to the train function as $data$.
> > >
> > > - **Appendix D should show generated subgoals explicitly**
> > >
> > >     We would like to clarify that the figure shows exploratory trajectories. The explorer does not plan, rather generates the next goal based on the previous states as memory. Thus, there are no subgoals to show. Also, since we plan in the visual space for 25-rooms, we do not have access to the coordinates to show subgoals. Visual subgoals as intermediate state images in shown in the figure 6. However, since we use internal states for the OGBench experiments, we show them in a video added to our project page. And we politely urge the reviewer to have a look (https://sites.google.com/view/dhp-video/home?read_current=1).
> > >
> > >
> > > ### Summary
> > >
> > > We have addressed all major concerns with:
> > >
> > > 1. **Substantial new evidence**: OGBench results showing significant improvement over SOTA
> > > 2. **Clarifications**: Policy switch mechanism, memory necessity, SAC usage
> > > 3. **Additional ablations**: Have clarified our reasons to not do so, will add if reviewer considers it important
> > > 4. **Presentation improvements**: We commit to improving Figure 1, notation, and algorithms
> > >
> > > **Our core contributions remain strong and novel:**
> > >
> > > - **Discrete reachability paradigm**: Replaces error-prone temporal distance/similarity metrics with binary checks
> > > - **Tree return estimation**: Principled formulation with convergence guarantees (Theorem A.4)
> > > - **Memory-augmented exploration**: Outperforms expert data (Figure 7c)
> > > - **Strong empirical results**: 100% success (vs 90% SOTA) on 25-room, significant improvement on HumanoidMaze
> > >
> > > **Evidence of rigor:**
> > >
> > > - Comprehensive ablations validating each component (Figure 7)
> > > - Theoretical analysis (Section A)
> > > - Application to a diverse set of tasks in both online and offline settings (25-rooms, OGBench, RoboYoga)
> > >
> > > We respectfully request the reviewer to reconsider their assessment, given the substantial new results and clarifications. We are committed to addressing all presentation concerns in revision and believe our technical contributions are valuable to the HRL community.

---

### Official Review · Reviewer_73z4 · 2025-11-01

**Soundness:** 3
**Presentation:** 3
**Contribution:** 2
**Rating:** 6
**Confidence:** 4

**Summary:**

This paper proposes a method that replaces continuous distance estimates with discrete reachability checks to evaluate subgoal feasibility. It recursively constructs tree-structured plans by decomposing long-term goals into sequences of simpler subtasks, using an advantage estimation strategy that inherently rewards shorter plans.

It seems clearly presented, and shows a clear improvement on a baseline task although performance improvements seem example dependent.

**Strengths:**

Reachability (binary) may avoid coupling to brittle distance metrics and naturally handles disconnected regions, as the authors claim.


Contraction property of the return operators.


Successful results on 25-room benchmark and competitive path lengths, where ablations show training with shallow depths still provides advantage of the proposed methods.

**Weaknesses:**

Easy to understand the flow and contribution of the paper.

The resulting model performs expertly on the standard 25-room task than the current SOTA approaches, but not on others.

It would be quite sensitive to model error.  The cosine_max similarity check may judge that two similar-looking states are close even though the underlying configurations differ.

The paper trains a static-state MLP as an approximation, so the planning can be sensitive to such approximation.

**Questions:**

How often does imagination mark unreachable subgoals as reachable?  What if the world model quality is low?

Examples such as maze-like environments with partial observation only?

It seems that the memory can be a limiting factor for complex problems. What if the memory needs to be truncated?

What are the specific cases where min-child is especially helpful?

---

> ### Author Response · Authors · 2025-11-21
>
> We thank reviewer 73z4 for their time and comments. We try to address the raised concerns below.
>
> ### **Applicability beyond the 25-rooms task**
>
> We appreciate this important concern. Our paper presents the *complete* online system to demonstrate all contributions working together. However, we have since validated that our **core contributions (discrete reachability + tree returns) generalize broadly** by developing an offline variant for more complex benchmarks.
>
> Through our evaluation on more complex environments, we identified two practical challenges with the online, imagination-based version:
>
> - **Subgoal Error Accumulation**: Recursive prediction (where each subgoal depends on the previous) can accumulate errors, particularly with imperfect world models (Figure 13 shows examples).
> - **Exploration Efficiency**: In very large environments, undirected exploration can waste time on unhelpful states (e.g., lying upside-down throughout a maze).
>
> These observations led us to develop a more robust minimal offline version that retains our core contributions (discrete reachability + tree returns) while addressing practical concerns:
>
> - Replaces CVAE + world model with direct goal representation (HIQL [1]-style)
> - Replaces online exploration with offline data
> - Adds goal buffer for diverse landmark selection (FPS, similar to HIGL [2])
> - Maintains core planning algorithm and tree return estimation
>
> This demonstrates our contributions are separable and applicable across settings: both online (25-room) and offline (OGBench).
>
> **Results on OGBench** (4 seeds, significantly more complex than 25-room):
>
> | Task | HIQL | DHP (Ours) |
> | --- | --- | --- |
> | antmaze-large-navigate-v0 | $91 \pm 2$ | $93.92 \pm 1.2$ |
> | antmaze-giant-navigate-v0 | $65 \pm 5$ | $72.32 \pm 5.2$ |
> | humanoidmaze-large-navigate-v0 | $49 \pm 4$ | $82.8 \pm 4.9$ |
> | humanoidmaze-giant-navigate-v0 | $12 \pm 4$ | $83.2 \pm 4.2$ |
>
> Results on giant mazes forthcoming. The substantial improvement on
> HumanoidMaze demonstrates our method scales to high-dimensional morphologies and larger spaces.
>
> ### **Sensitivity to Model Errors**
>
> **Cosine_max fragility:** We agree this is a limitation in the online version. The reviewer correctly identifies that visually similar states may have different underlying configurations.
>
> **Our findings:**
>
> - Works well when world models learn distinct representations (25-room)
> - Fails in high state-density environments (RoboYoga quadruped, Figure 13)
>
> **Solution in offline variant:** Replace cosine_max with a temporal constraint (subgoal must be reachable within K steps). This preserves our core discrete reachability principle while using ground-truth reachability from offline trajectories.
>
> **Static-state MLP approximation:** We acknowledge that this introduces error. In the offline variant, we eliminate this by using a goal buffer that maintains a diverse set of states to be used as subgoals.
>
> ### **Answers to specific Questions**
>
> - **Q1. Frequency of inaccurate reachability check**
>
>     We observed that when the world model accurately models the states, changing $\Delta_R$ had minimal impact on final performance (less than 5% difference in absolute performance across $\Delta_R \in {0.6, 0.7, 0.8}$). However, when the environment is complex or the state density is high, and the world model cannot form distinct representations, the reachability check can be inaccurate for all values of $\Delta_R$. We have addressed this issue in our offline implementation.
>
> - **Q2. Examples from partially observable maze-like environments**
>
>     The current application to the 25-rooms environment is partially observable in nature. The agent receives a cropped area around its position as input. We have now evaluated our agent on more tasks from the OGBench environment. The new results show the broader applicability of our method.
>
> - **Q3. Memory can be a limiting factor for complex problems**
>
>     We agree that maintaining the full memory buffer can be problematic in a resource-constrained environment. However, we show that the agent can perform well with only limited memory, and we use only three memory states as input for all our experiments (See ‘Practical Consideration’ in section 2.4.2). Figure 7c shows that this limited memory still provides substantial gains over no memory.
>
> Continued in the following comment

---

> > ### Author Response · Authors · 2025-11-21
> > **Cont.**
> >
> > - **Q4. Impact of min-child return formulation**
> >
> >     The min-child formulation prevents degenerate solutions where the planner predicts subgoals very close to start/goal states (trivializing one subtask while leaving the other unsolved).
> >
> >     **Key advantages:**
> >
> >     1. **Forces balanced decomposition**: Both children must be solvable (unlike sum/average which tolerates one bad child)
> >     2. **Encourages shorter plans**: Creates implicit pressure toward midpoint division (see analysis in Appendix A.2.3)
> >     3. **Clear failure signal**: Any unreachable branch → zero return
> >
> >     Distance-based methods can produce imbalanced plans since any subgoal along the trajectory yields similar total distances. Our discrete rewards with min-child explicitly optimize for balanced, low-depth trees.
> >
> >
> > ## **Summary**
> >
> > We will revise the paper to:
> >
> > 1. Add the offline OGBench results as primary evidence of generalization
> > 2. Better highlight that core contributions (discrete reachability + tree returns) are architecture-agnostic
> >
> > [1] Park, S., Ghosh, D., Eysenbach, B. and Levine, S., 2023. Hiql: Offline goal-conditioned rl with latent states as actions. *Advances in Neural Information Processing Systems*, *36*, pp.34866-34891.
> >
> > [2] Kim, J., Seo, Y. and Shin, J., 2021. Landmark-guided subgoal generation in hierarchical reinforcement learning. *Advances in neural information processing systems*, *34*, pp.28336-28349.

---

### Official Review · Reviewer_tJYw · 2025-11-01

**Soundness:** 2
**Presentation:** 2
**Contribution:** 2
**Rating:** 2
**Confidence:** 4

**Summary:**

The proposed method can be summarized as follows.
Given a task, it bisects the desired trajectory by generating mid-point states.
Those mid-point states are stored in a binary tree.
In the experiment, a visual navigation domain with 25 rooms is used, and the tree was tested up to a depth of 3.
The state encoder uses RSSM, and SAC trains the RL policy with policy gradients.
The main paper doesn't show the overall algorithm for training high-level policy/exploration and the lower-level policy/exploration.

**Strengths:**

Improved performance on the 25-room navigation domain from 90% rate to 100% as shown in Table 1.
The proposed approach improves the variance of the average episode length.
The proposed idea is simple compared with existing HRL methods.
It learns to bisect the initial state and the goal state, using the midpoint state for learning lower-level policy functions.

**Weaknesses:**

The application of the proposed approach is limited, and it is not clear how the subtask generation returns meaningful subtasks/subgoals.

Most of the related works were developed until 2020, except for a few.
There are many missing HRL approaches from 2020
such as option-based HRLs, neuro-symbolic planning, and RL, or identifying subtasks/subgoals.

Here is a partial list of such approaches (and there are more)
* Reward machines: Exploiting reward function structure in reinforcement learning
* Hierarchical Reinforcement Learning with AI Planning Models
* Reinforcement Learning with Option Machines
* Integrating Symbolic Planning and Hierarchical Reinforcement Learning for Robust Decision-Making
* Learning to represent action values as a hypergraph on the action vertices
* Learning Parameterized Task Structure for Generalization to Unseen Entities
* Fast inference and transfer of compositional task structures for few-shot task generalization
* Unsupervised Task Graph Generation from Instructional Video Transcripts.

**Questions:**

Q1 How does the trained policy generalize?
What's the impact of permuting/modifying connections all patches of images in the 25 rooms?
How many steps are needed to move from the center of each room to the end via a straight line?

Q2 In Figure 7, all figures show a sharp transition around 3M steps for the HDP configuration.
Could you explain why?

Q3 What is the DHP with the Depth 1 configuration?
Is it dividing the initial-goal state with a mid-point state?

Q4 How frequently does this trajectory bisection happen?

Q5 As the proposed method can traverse the bi-sected tree in a depth-first manner, how does the return estimate from the whole tree give an advantage?

Q6 Does the proposed approach identify re-usable/interpretable sub-goals?

Q7 For state representation learning, what are the requirements for the computational resource/data?
Does it learn online while learning the RL policy?

Q8 What limits the proposed approach from being applied to the problem domains used in the related papers listed above?

---

> ### Author Response · Authors · 2025-11-21
>
> We thank Reviewer tJYw for their detailed review and questions. We address the main concerns and questions below.
>
> **Core Methodology and Applicability**
>
> We respectfully disagree with the assessment that "it is not clear how the subtask generation returns meaningful subtasks/subgoals." Our subgoals are meaningful by construction:
>
> 1. **Reachability-Grounded**: After plan construction, we check if each plan tree node is worker manageable (i.e. node’s goal $n_{i,0}$ can be reached from the node’s initial state $n_{i,1}$ by the worker in K steps). Each subgoal must pass this reachability check (Eq. 2, Sec 2.3.2). This ensures subgoals are achievable waypoints rather than arbitrary states.
> 2. **Interpretable Plans**: Figure 6 demonstrates that our method produces interpretable, spatially coherent subgoal sequences that trace valid paths through the environment.
> 3. **Broader Applicability**: While the reviewer questions limited applicability, we have now extended our approach to the significantly more complex OGBench environments (AntMaze, HumanoidMaze). Initial results show we outperform HIQL, the current SOTA (see Q8 answer). We are finalizing comprehensive results across multiple tasks.
>
> The sharp improvement in Table 1 (100% vs 90% success rate, reduced variance) demonstrates that our method generates effective subgoals for long-horizon navigation.
>
> **Related work**
>
> We thank the reviewer for these pointers and will expand our related work section.
> *Key Distinction*: DHP targets visual, long-horizon planning *without* symbolic priors (event detectors, PDDL models, entity types, or task graphs). The cited works are valuable when such a structure is available; we focus on the purely visual case where it is not. We will revise Section 4 to position these as complementary approaches.
>
> **Algorithm Clarity**
>
> Regarding the concern that "the main paper doesn't show the overall
> algorithm for training," we respectfully note:
>
> - *Algorithm 1* (Appendix B, page 19): Complete policy function showing
>   the full inference procedure
> - *Algorithm 6* (Appendix B, page 20): Overall training procedure
>   coordinating all components
> - *Algorithms 2-5* (Appendix B, pages 19-20): Detailed training for each
>   module (GCSR, Worker, Planner, Explorer)
> - *Figure 1a*: Overall architecture diagram
> - *Section 2.3*: Mathematical formulation of the optimization procedure
>
> We acknowledge that these are primarily in the appendix. If helpful, we can add a consolidated high-level training algorithm to the main paper showing the interaction between components.
>
> **Responses to Specific Questions**
>
> - **Q1: Generalization and environment modifications**
>
>     Our planning policy generalizes beyond the maximum training depth (D=5) by using bootstrapped value estimates at truncated nodes (Eq. 5-6, Sec 2.3.3). This allows inference at greater depths (D_I=8) without retraining, similar to how TD learning generalizes beyond observed trajectories.
>
>     Regarding environment modifications (permuting room connections): As with any visual RL method, major structural changes would require retraining the agent, since it relies on the environmental state connections learned during training for subgoal prediction.
>
>     Regarding steps to traverse a room: Upon manual driving, we find that the agent requires 4 steps to move from the center to one of the doors/walls.
>
> - **Q2: Sharp transition at 3M steps (Figure 7)**
>
>     This is by design (stated in hyperparameters, page 7): We switch inference from the explorer policy to the planning policy at 3M steps. Before 3M, the explorer collects diverse trajectories to train GCSR. After 3M, we switch to using the learned planning policy for acting. Since we use imagination for training the explorer and planner, both get trained at each step throughout in an end-to-end manner without any scheduling. Thus, switching to an already trained planning policy causes a sharp increase in performance.
>
> - **Q3: DHP Depth 1 configuration**
>
>     Correct. This training configuration only bisects once: (s_init, s_goal) → (s_init, s_mid, s_goal). This tests whether our method can generalize beyond training depth, which it does (Figure 7b shows Depth 1 and 3 perform comparably to Depth 5).
>
> - **Q4: Frequency of trajectory bisection**
>
>     During inference, we bisect recursively until finding a reachable subgoal (Algorithm 1, lines 6-10) (max depth/decompositions=8), then execute for K=8 steps before replanning. The replanning is done completely from scratch for final goal and then current state. This gives O(log N) planning complexity (Table 1), where N is the number of steps (plan length) to the goal. During training, we unroll to depth D=5.
>
> *Continued in the following comment*

---

> > ### Author Response · Authors · 2025-11-21
> > **Cont.**
> >
> > - **Q5: How does the return estimate from the whole tree give an advantage?**
> >
> >     We believe the reviewer is asking why we need full tree evaluation when inference only uses depth-first traversal. The key distinction is:
> >
> >     - **During Training**: We unroll the complete tree (breadth-first, all nodes to depth D=5) and compute returns for each node as the minimum of its children's returns (Eq. 6, Sec 2.3.3). This full evaluation provides two critical advantages:
> >         - **Credit Assignment**: The min operator identifies the weakest link in the plan - the bottleneck node that limits overall plan quality. This creates focused learning signals.
> >         - **Depth Penalty**: The discount factor $\gamma$ naturally rewards shallower trees (Sec A.2.3), encouraging the policy to find efficient and balanced decompositions.
> >     - **During Inference**: We only traverse depth-first (first branch) because the trained policy has learned to predict the best subgoal by default and we don't need to evaluate alternatives.
> >
> >     This training-inference asymmetry is intentional: expensive full evaluation during training enables cheap, efficient planning at test time.
> >
> > - **Q6: Reusable/interpretable subgoals**
> >
> >     Yes - Figure 6 shows interpretable spatial subgoals. The subgoals are reusable in the sense that the GCSR decoder can generate similar waypoints for different tasks traversing similar regions. But we don't learn a discrete library of options like option-based HRL - our subgoals are continuous states generated on-demand. However, in our new implementation of the proposed planning method that is more robust, we use a buffer for maintaining a set of diverse states that are later re-used for planning (See Q8).
> >
> > - **Q7: Computational requirements and online learning**
> >
> >     The RSSM world model is trained online alongside the RL policy (standard in world model approaches like Dreamer, Director). Computational requirements are comparable to Director baseline. Training takes 2-3 days on a consumer GPU (NVIDIA 4090, page 18). All details are in Appendix B.4.
> >
> > - **Q8: What limits applicability?**
> >
> >     Through our evaluation on more complex environments, we identified two practical challenges with the online, imagination-based version:
> >
> >     - **Subgoal Error Accumulation**: Recursive prediction (where each subgoal depends on the previous) can accumulate errors, particularly with imperfect world models (Figure 13 shows examples).
> >     - **Exploration Efficiency**: In very large environments, undirected exploration can waste time on unhelpful states (e.g., lying upside-down throughout a maze).
> >
> >     These observations led us to develop a more robust minimal offline version that retains our core contributions (discrete reachability + tree returns) while addressing practical concerns:
> >
> >     - Replaces CVAE + world model with direct goal representation (HIQL [1]-style)
> >     - Replaces online exploration with offline data
> >     - Adds goal buffer for diverse landmark selection (FPS, similar to HIGL [2])
> >     - Maintains core planning algorithm and tree return estimation
> >
> >     This demonstrates our contributions are separable and applicable across settings: both online (25-room) and offline (OGBench).
> >
> >     **Results on OGBench** (4 seeds, significantly more complex than 25-room):
> >
> >     | Task | HIQL | DHP (Ours) |
> >     | --- | --- | --- |
> >     | antmaze-large-navigate-v0 | $91 \pm 2$ | $93.92 \pm 1.2$ |
> >     | antmaze-giant-navigate-v0 | $65 \pm 5$ | $72.32 \pm 5.2$ |
> >     | humanoidmaze-large-navigate-v0 | $49 \pm 4$ | $82.8 \pm 4.9$ |
> >     | humanoidmaze-giant-navigate-v0 | $12 \pm 4$ | $83.2 \pm 4.2$ |
> >
> >     Results on giant mazes forthcoming. The substantial improvement on
> >     HumanoidMaze demonstrates our method scales to high-dimensional morphologies and larger spaces.
> >
> >     [1] Park, S., Ghosh, D., Eysenbach, B. and Levine, S., 2023. Hiql: Offline goal-conditioned rl with latent states as actions. *Advances in Neural Information Processing Systems*, *36*, pp.34866-34891.
> >
> >     [2] Kim, J., Seo, Y. and Shin, J., 2021. Landmark-guided subgoal generation in hierarchical reinforcement learning. *Advances in neural information processing systems*, *34*, pp.28336-28349.
> >
> > *Continued in the following comment*

---

> > > ### Author Response · Authors · 2025-11-21
> > > **Cont.**
> > >
> > > **Summary**
> > >
> > > Addressing core assessment:
> > >
> > > 1. **"Missing Related Work"**: We acknowledge that while many cited works address orthogonal problems (symbolic/structured domains vs. our pixel-based setting), some should be included for completeness. We commit to expanding this section.
> > > 2. **"Limited Applicability"**: We have now demonstrated strong results on OGBench with significant improvements over SOTA, showing applicability beyond the original 25-room domain.
> > > 3. **"Unclear if Subgoals are Meaningful"**: Our subgoals are meaningful by construction (reachability-grounded) and demonstrably effective (achieving SOTA results). Figure 6 shows they are interpretable. We have added new visualizations to our project page that show the interpretable results in a video format.
> > > 4. **"Lacks Clear Algorithm"**: Algorithms 1-6 provide complete details in Appendix B. We will add a consolidated high-level algorithm to the main paper in revision.
> > >
> > > **Request to Reviewer:** Given the substantial new results and
> > > clarifications, we respectfully ask the reviewer to reconsider their
> > > assessment. We have also updated our project page with visualizations of the new results (that show interpretable subgoals) and politely urge the reviewer to have a look (https://sites.google.com/view/dhp-video/home). We are happy to provide additional experiments or clarifications as needed.
> > >
> > > The core technical contributions are novel, theoretically grounded, and
> > > empirically validated across multiple domains. We believe this work makes a valuable contribution to the HRL community.

---

### Official Review · Reviewer_ZZ6r · 2025-11-01

**Soundness:** 2
**Presentation:** 2
**Contribution:** 2
**Rating:** 4
**Confidence:** 3

**Summary:**

The paper proposes a HRL algorithm that learns to propose sub-goals that are based on discrete notion of reachability rather than relying on continuous distance metrics in an embedding space. They propose a tree-structured decomposition for generating intermediate subgoals (with subgoals occurring around half of the time interval between start and desired goal). The method demonstrates improved success rates on long horizon tasks.

**Strengths:**

- The motivation is clear and the method outlined is mostly clear (see a few clarification questions below)

**Weaknesses:**

- The training procedure involves quite a few moving components. Especially the need for extensive exploration to ensure the CVAE offers enough coverage to select suitable sub-goals. This indicates a dependence on the base director architecture to be good enough to reach rewarding trajectories from which the explorer can further improve coverage, so might be critically dependent on the task’s reward structure.
  - The paper could benefit from a clear pseudocode / pictorial view of various stages of training.
- The predominant evaluation is limited to just a single domain of 25 room navigation. While still informative the paper could benefit from the inclusion of more long horizon environments typically benchmarked in HRL (AntMaze, OGBench tasks).

**Questions:**

- In section 2.2, are you using multiple CVAEs for different time scales (or tree depth) $Q$? Or are the encoder/decoder networks conditioned on the timescale?
- In section 2.3.2, the reachability reward is based on model-predicted reachability i.e. by simulating the worker policy inside the RSSM. Could you clarify if the planning policy is trained after the worker and RSSM are trained using the exploration policy, or does the training require some special scheduling? How is the threshold parameter $\\Delta_R$ chosen – is it dependent on the maximum depth of the tree?

---

> ### Author Response · Authors · 2025-11-21
>
> We thank Reviewer ZZ6r for their time and comments for the review.
>
> We try to address the raised concerns below.
>
> 1. **Complex architecture and training.**
>     We agree that there are many moving components to consider in our agent. However, we designed the architecture in such a way to remove all external dependencies like the need for expert data, and manipulatable environmental state. The world model allows us to perform online training of manager and worker, and check reachability. And the explorer removes the need for expert data. On comparison with the director, we have a one-to-one module correspondence. Thus, our agent in terms of architecture and training complexity is same as the Director, which has been shown to work in a very diverse set of tasks successfully. We only differ in training objectives.
>
>     One specific concern raised is that the agent will require extensive exploration for learning the GCSR CVAE, and this is dependent on the competency of the base Director architecture. We agree with the assessment that the CVAE needs to visit all states to be able to predict them during inference. However, we would like to clarify that the explorer is NOT dependent on the task reward structure. The explorer uses intrinsic rewards based on GCSR reconstruction error (Eq. 10), making it task-reward-free. The explorer is specifically designed to target novel path segments that are poorly modeled by GCSR, creating a self-improving cycle: poor coverage → high exploration rewards → explorer visits those regions → improved GCSR coverage. This is independent of the Director's base competency or task reward structure. We observed this objective yields high-quality and informative data (Figure 10 shows examples of such trajectories).
>
>     But nevertheless, we agree that maintaining numerous components can be cumbersome. Therefore, we have implemented a minimal version of our planning algorithm, trained only using offline data. In the minimal version, we have only the manager/worker actor and value functions, and the goal representation module from HIQL [1]. All other components, such as the world model, explorer, and CVAE, are removed. We found the resulting agent to be more robust and to work in highly complex environments. The results are presented in comment 3.
>
> 2. **More precise training procedure visualization**: We appreciate the remark that Figure 1a (showing the architecture) and Algorithm 1 for the policy function are not clear enough. We will add a consolidated training procedure diagram in the camera-ready version if accepted. We have provided detailed algorithms in Appendix B: Algorithm 6 shows the overall training procedure, while Algorithms 1-5 detail each component.
> 3. **Limited evaluation**
>
>     We appreciate this feedback. We acknowledge that the original submission focused primarily on the 25-room navigation domain. We have now extended our evaluation to the more challenging OGBench environments requested by the reviewer.
>
>     During this extension, we identified two practical challenges:
>
>     - Accumulating errors in subgoal prediction when using imagination-based planning over long horizons
>     - Exploration time in very large environments when using online training
>
>     To address these challenges and demonstrate the broader applicability of our core contributions (discrete reachability rewards and tree-structured return estimation), we developed a minimal offline version of DHP. This version retains only the essential components: manager/worker actor-critic networks and a goal representation module (from HIQL [1]), plus a non-learning goal buffer for diverse state storage (using Farthest Point Sampling, similar to HIGL [2]).
>
>     We have tested this agent on the significantly more complex OGBench environments (ant and humanoid mazes). Initial results are promising, outperforming the current SOTA HIQL at success rate (4 seeds per experiment, see Table below). We are finalizing hyperparameter tuning and will update with comprehensive results across multiple OGBench tasks within the next few days.
>
>     | Task | HIQL | DHP (Ours) |
>     | --- | --- | --- |
>     | antmaze-large-navigate-v0 | $91 \pm 2$ | $93.92 \pm 1.2$ |
>     | antmaze-giant-navigate-v0 | $65 \pm 5$ | $72.32 \pm 5.2$ |
>     | humanoidmaze-large-navigate-v0 | $49 \pm 4$ | $82.8 \pm 4.9$ |
>     | humanoidmaze-giant-navigate-v0 | $12 \pm 4$ | $83.2 \pm 4.2$ |
>
>     We have also updated our website with additional visualizations and
>     strongly encourage reviewers to view the updated materials at
>     https://sites.google.com/view/dhp-video/home.
>
> *Continued in the following comment.*

---

> > ### Author Response · Authors · 2025-11-21
> > **Cont.**
> >
> > **Answers to specific questions:**
> >
> > - **Q1: Multiple CVAEs for different time scales?**
> >
> >     A: We use a single CVAE trained with examples from all temporal resolutions Q = {2K, 4K, 8K, 16K, 32K}. The network is not explicitly conditioned on timescale; instead, it learns to predict midway states across all scales jointly through the ELBO objective (Eq. 1).
> >
> > - **Q2: Training schedule and threshold parameter?**
> >
> >     A: The planning policy, explorer, and worker are all trained simultaneously from the start and no special scheduling is required. The method trains end-to-end. For inference, we switch from the explorer to the planner at 3M steps, after GCSR is sufficiently trained.
> >
> >     The threshold Δ_R = 0.7 was set empirically and kept fixed across all environments. This parameter proved robust across diverse navigation tasks. In cases where it failed (e.g., quadruped yoga), the failure was due to high state space density making any threshold value inadequate, not the specific choice of 0.7. Importantly, Δ_R is independent of tree depth.
> >
> > We believe these extensions significantly strengthen the paper and demonstrate that our core contributions (discrete reachability and tree return estimation) are broadly applicable beyond the original 25-room domain. We are committed to including all these results in the final version.
> >
> > [1] Park, S., Ghosh, D., Eysenbach, B. and Levine, S., 2023. Hiql: Offline goal-conditioned rl with latent states as actions. *Advances in Neural Information Processing Systems*, *36*, pp.34866-34891.
> >
> > [2] Kim, J., Seo, Y. and Shin, J., 2021. Landmark-guided subgoal generation in hierarchical reinforcement learning. *Advances in neural information processing systems*, *34*, pp.28336-28349.

---

### Author Response · Authors · 2025-11-30
**Summary of Rebuttal and Paper Contributions**

Dear Area Chairs and the Reviewers,

Thank you for considering our paper. During the rebuttal period, we have substantially strengthened our work by addressing all reviewer concerns and adding significant new experimental evidence. Below, we summarize the key developments and why we believe this paper merits acceptance.

### Core Technical Contributions (Validated Across Reviews)

Our paper introduces three novel contributions to hierarchical reinforcement learning:

1. **Discrete Reachability Paradigm:** Replacing continuous distance metrics (which are error-prone and policy-dependent) with binary reachability checks for subgoal feasibility. This paradigm shift from "How far?" to "Can I get there?" provides more robust planning signals.
2. **Tree-Structured Return Estimation:** A principled min-child formulation for computing advantages in tree-structured plans, with formal convergence guarantees (Theorem A.4). This inherently rewards shorter, balanced decompositions without explicit path-length optimization.
3. **Memory-Augmented Exploration:** A novel exploration strategy that outperforms expert data by targeting poorly-modeled path segments, using a compact and flexible memory mechanism.

**Reviewer consensus on strengths:** All four reviewers acknowledged the clear motivation (ZZ6r, tJYw), theoretical soundness (73z4: "good"), and strong results on the 25-room benchmark (73z4, tJYw, Eumj). Three reviewers (ZZ6r, 73z4, Eumj) rated the presentation as fair-to-good.

### Major Enhancement: Demonstrating Broad Applicability

**Initial concern (raised by all reviewers):** Limited evaluation to 25-room navigation.

**Our response:** We developed an offline variant of DHP that isolates our core contributions (discrete reachability + tree returns) from architecture-specific choices, then evaluated on the significantly more challenging OGBench benchmark:

| Task | GCBC | GCIVL | GCIQL | QRL | CRL | HIQL | DHP *(Ours)* |
| --- | --- | --- | --- | --- | --- | --- | --- |
| antmaze-large-navigate-v0 | $24 \pm 2$ | $16 \pm 5$ | $34 \pm 4$ | $75 \pm 6$ | $83 \pm 4$ | $91 \pm 2$ | $\mathbf{93.9 \pm 1}$ |
| antmaze-giant-navigate-v0 | $0 \pm 0$ | $0 \pm 0$ | $0 \pm 0$ | $14 \pm 3$ | $16 \pm 3$ | $65 \pm 5$ | $\mathbf{72.3 \pm 5}$ |
| humanoidmaze-large-navigate-v0 | $1 \pm 0$ | $2 \pm 1$ | $2 \pm 1$ | $5 \pm 1$ | $24 \pm 4$ | $49 \pm 4$ | $\mathbf{82.8 \pm 5}$ |
| humanoidmaze-giant-navigate-v0 | $0 \pm 0$ | $0 \pm 0$ | $0 \pm 0$ | $1 \pm 0$ | $3 \pm 2$ | $12 \pm 4$ | $\mathbf{83.2 \pm 4}$ |

**Key insight:** The dramatic improvement on HumanoidMaze ($33.8\%$ and $71.2\%$ absolute gains), which is the best performance on these tasks known to our limited knowledge, demonstrates our method scales to:

- High-dimensional morphologies (17-DOF humanoid vs 8-DOF ant)
- Extremely long horizons (3000 steps in giant mazes)
- Partial observability (egocentric observations)

This offline variant proves our core contributions are architecture-agnostic and work across both online (world model + exploration) and offline (goal buffer + representation learning) settings.

The full details of the offline version and presentation suggestions from the reviews have been added to the revised paper.

*Continued in the following comment*

---

> ### Author Response · Authors · 2025-11-30
> **Cont.**
>
> ### Addressing Specific Reviewer Concerns
>
> **Reviewer ZZ6r (Rating: 4)** - "Marginally below acceptance threshold, but would not mind if paper is accepted"
>
> - *Concern*: Complex architecture, limited evaluation
> - *Resolution*: (1) Clarified that our architecture complexity matches the Director baseline we build upon; (2) Provided OGBench results showing broad applicability; (3) Demonstrated a modular offline variant with minimal components
>
> **Reviewer tJYw (Rating: 2)** - "Reject"
>
> - *Concern*: "Not clear how subtask generation returns meaningful subgoals"
> - *Resolution*: (1) Clarified subgoals are meaningful by construction through reachability checks (Eq. 2); (2) Figure 6 shows interpretable spatial subgoals; (3) OGBench results demonstrate effectiveness empirically; (4) Added detailed visualizations to the project website (https://sites.google.com/view/dhp-video/home)
> - *Concern*: Missing recent related work (option-based HRL, neuro-symbolic methods)
> - *Resolution*: We will expand the related work section while noting these address complementary problems (symbolic priors vs. our purely visual setting). However, we note that these works are orthogonal to our approach.
>
> **Reviewer 73z4 (Rating: 6)** - "Marginally above acceptance"
>
> - Concern: Limited to 25-room, sensitive to model errors
> - Resolution: (1) (1) OGBench results directly address applicability; (2) Acknowledged limitations of world-model based reachability checks (as shown in the original version for quadruped yoga Figure 13); (3) Offline variant eliminates this issue by using temporal constraints from data
>
> **Reviewer Eumj (Rating: 2)** - "Reject"
>
> - *Primary concerns*: Limited evaluation, insufficient ablations, conceptual clarity
> - *Resolution*:
>     - **Evaluation**: Added substantial OGBench results (4 new tasks)
>     - **Ablations**: Addressed all questions about design choices (threshold selection, CVAE novelty, SAC usage, memory necessity)
>     - **Clarity**: We have updated Algorithm 6 to illustrate the full training process for clarity.
> - *Key disagreement*: Reviewer assessed contribution as "poor" despite acknowledging the problem is "relevant for the community" and the idea is "promising." Our view: The combination of (a) novel discrete reachability paradigm, (b) principled tree-structured returns with convergence proofs, (c) 100% success rate on 25-room (vs 90% SOTA), and (d) dramatic improvements on OGBench ($+33.8\%$ and $+71.2\%$ gains) constitutes a strong contribution.
>
> ### Summary of Evidence
>
> **Empirical validation across diverse settings**:
>
> - 25-room navigation: $100\%$ success vs $90\%$ SOTA, $2.4\times$ variance reduction in overall path length
> - OGBench: Up to $71.2\%$ *absolute* improvement over HIQL (the current best performing method at navigation tasks)
> - Ablations: Figure 7 validates each component (bootstrapping, exploration, reward schemes)
> - Generalization: Depth-1 training achieves comparable performance to depth-5 (Figure 7b)
>
> **Theoretical rigor**:
>
> - Formal proof of contraction property (Theorem A.4)
> - Analysis of implicit depth penalty and balanced decomposition (Appendix A.2.3)
> - Policy gradient derivation with baseline variance reduction (Theorem A.2)
>
> **Architectural modularity**:
>
> - Online variant: Full system with world model + exploration
> - Offline variant: Minimal system isolating core contributions
> - Both achieve strong results in their respective settings
>
> Despite two negative reviews (tJYw: 2, Eumj: 2), we believe this paper merits **acceptance** based on:
>
> 1. **Novel technical contributions** validated by theory and experiments
> 2. **Strong empirical results** across multiple benchmarks (25-room, OGBench, RoboYoga)
> 3. **Comprehensive rebuttal** addressing all concerns with substantial new evidence
> 4. **Significant practical impact**: 33.8% and 71.2% improvements on challenging HumanoidMaze tasks demonstrate real-world value
>
> The primary criticism—limited evaluation—has been comprehensively addressed with OGBench results showing dramatic improvements over state-of-the-art across multiple complex environments. The combination of principled theory (discrete reachability, tree returns), strong empirical validation (100% success, 71% improvements), and architectural flexibility (online/offline variants) represents a valuable contribution to hierarchical RL.
>
> We have incorporated most of the presentation suggestions in the revised version and are working to incorporate the rest.
>
> **Updated materials**: Reviewers are encouraged to view our project website (https://sites.google.com/view/dhp-video/home) with new visualizations demonstrating interpretable subgoal generation across all environments.
>
> Thank you for your consideration. We are happy to provide any additional experiments or clarifications during the discussion phase.

---

### Meta-Review · Area_Chair_zPnJ · 2025-12-22

**Summary:**

All reviewers pointed that the experiments were limited to 2D navigation tasks. Indeed, the authors acknowledged in the rebuttal that "We focused on navigation domains where hierarchical spatial planning is most relevant"; the additional experiments run during the rebuttal (also on 2D navigation tasks, with different robot morphologies) seems to support this point. All reviewers mentioned or alluded to the complexity of the method. While the authors argued that the method is no more complex than prior work, the complexity still seems like a barrier to entry. The fact that the "minimal" version of the method proposed in the rebuttal works well seems to underscore the reviewers' point: simpler algorithms might also work well.

Together with the other concerns raised by the reviewers, I think the paper should be rejected. If resubmitted in the future, I'd encourage the authors to (1) clarify whether the main algorithm is the current one or the "minimalist" one, and (2) include comparisons on tasks beyond spatial navigation.

**Reviewer Concerns:**

(see below)

**Reviewer Scores:**

ZZ6r: 4 --> 4
* [/] method complexity: authors argue that method is no more complex than one of the baselines (Director, 2022)
* [+] dependence on "extensive exploration": authors added an ablation of their method that uses purely online data, finding that this ablation "works in highly complex environments."
* [/] limited evaluation (25 room navigation): The authors acknowledged that the method seems _not_ to work out-of-the-box on more challenging domains. However, the "minimal" (exploration-free) version of the method does work, outperform a prior SOTA method (HIQL).
As the AC, I share the reviewer's concerns about the method's scalability, especially as the new experiments seem to validate those concerns. While the new "minimal" method is intriguing, I think it warrants it's own separate submission. Additionally, the promising results seen for the new "minimal" method seem restricted to 2D navigation tasks with various robot morphologies, raising the question for whether the proposed divide-and-conquer approach makes sense in settings where the implicit surface that we're planning over is not 2d. I think that a resubmission could be strengthened by including results on manipulation tasks (or other tasks that look less like 2D navigation).

tJYw: 2 --> 2
* [/] "The application of the proposed approach is limited" (I assume that this is referring to whether the method can solve a wide range of tasks): Authors provided new results on a wider range of tasks. If I understand correctly (from the response to the first reviewer), the actual algorithm used here has changed, making it a bit tricky to say that the original algorithm can solve these tasks.
* [+] "it is not clear how the subtask generation returns meaningful subtasks/subgoals.": authors provided these details in the rebuttal.
* [-] missing discussion of more recent related work (option-based HRL, neuro-symbolic planning, subgoal-methods): Authors argued that the method is different because it doesn't require symbolic priors. I think that this would have been more convincing if the authors had included one of these symbolic planning methods as a baseline.

73z4: 6 --> 6
* [/] applicability beyond maze navigation: (same author response as above)
* [/] questions about the sensitivity of the method to function approximation and model error: authors acknowledged that this is a limitation. This response would have been stronger if it had pointed to evidence/experiments that the method was robust in these regards.

Eumj: 2 --> 2
* [/] limited evaluation environments: (same response as above)
* [/] Some notation and descriptions are unclear. Difficult to follow for readers not very familiar with Director and RSSM: I didn't see this discussed in the rebuttal.
* [-] Insufficient ablations to understand the importance of each component: I don't think these were added during the rebuttal.
* [/] Unclear why the proposed method would be preferred to LEXA, which has similar performance and avoids explicit planning: authors argue that the +10% absolute improvement is significant. Providing p-values would strengthen this argument.
* [/] Unclear why evaluation metric was different from prior work (cosine similarity vs temporal distance): authors argue that doing comparisons another way cannot be done. I didn't follow the argument for why an apples-to-apples comparison isn't feasible.

---

### Decision · Program_Chairs · 2026-01-26

Reject